evolution, genetics, theoretical biology

cytoplasmic inheritance, intragenomic conflict, overlapping generations, sex chromosome, sexual conflict, soft selection

**Author for correspondence:**
Thomas J. Hitchcock
e-mail: th76@st-andrews.ac.uk

# Sex-biased demography modulates male harm across the genome

Thomas J. Hitchcock and Andy Gardner

School of Biology, University of St Andrews, St Andrews KY16 9TH, UK

 TJH, 0000-0002-6378-5023; AG, 0000-0002-1304-3734

Recent years have seen an explosion of theoretical and empirical interest in the role that kin selection plays in shaping patterns of sexual conflict, with a particular focus on male harming traits. However, this work has focused solely on autosomal genes, and as such it remains unclear how demography modulates the evolution of male harm loci occurring in other portions of the genome, such as sex chromosomes and cytoplasmic elements. To investigate this, we extend existing models of sexual conflict for application to these different modes of inheritance. We first analyse the general case, revealing how sex-specific relatedness, reproductive value and the intensity of local competition combine to determine the potential for male harm. We then analyse a series of demographically explicit models, to assess how dispersal, overlapping generations, reproductive skew and the mechanism of population regulation affect sexual conflict across the genome, and drive conflict between nuclear and cytoplasmic genes. We then explore the effects of sex biases in these demographic parameters, showing how they may drive further conflicts between autosomes and sex chromosomes. Finally, we outline how different crossing schemes may be used to identify signatures of these intragenomic conflicts.

## 1. Introduction

Sexual conflict [1–4] and kin selection [5–9] represent central strands of evolutionary biology, and recent years have seen an explosion of interest in the connection and interplay between the two [10–30]. Much of the theoretical attention that has been devoted to this topic has focused on how the incentive for male harm (i.e. traits that increase a male's mating success at the expense of the females with whom he interacts) may be curbed by relatedness between mates and between mate competitors, in a range of ecologically and demographically explicit population settings [24–29]. These theoretical predictions have motivated a growing body of empirical work on a diversity of organisms including flies [12–15,30], chickens [16], mites [17,18] and beetles [19–21,23].

However, this body of theory has focused on autosomal inheritance and has not considered how ecological and demographic factors shape sexual conflict across the rest of the genome. With recent interest in the evolution of sexual conflict on sex chromosomes [31] and an improving understanding of the molecular basis of harming traits [32], extending this theory to consider how kin selection may differentially mould male harm with respect to non-autosomal portions of the genome is crucial for guiding and interpreting future empirical work. In addition to providing an array of further comparative predictions for populations that differ with respect to ecological and demographic factors, this new theory would also yield comparative predictions at an intragenomic level, which is a particularly powerful approach as within-individual comparisons naturally control for variation in a diversity of confounding factors [33].

To investigate this, we adapt previous models of sexual conflict [24,26,28] for application to autosomal, sex chromosomal and cytoplasmic inheritance. Our analysis encompasses both male (XY and XO) and female (ZW and ZO) heterogametic systems, and therefore we investigate the possibility of male harm loci

occurring on X, Y and Z chromosomes. With respect to cytoplasmic factors, we consider the full range from strictly maternal to strictly paternal inheritance. We first provide a general overview, showing how sex-specific relatedness, reproductive value and the intensity of kin competition combine to determine the potential for male harm. We next analyse a series of ecologically, demographically and genetically explicit models [34,35], revealing how dispersal, overlapping generations, reproductive skew and the mechanism of population regulation modulate sexual conflict across different parts of the genome, and ignite intragenomic conflicts between nuclear and cytoplasmic genes. We then explore the effects of sex biases in these demographic factors, showing how they may drive further intragenomic conflicts between autosomes and sex chromosomes. Finally, we discuss how these theoretical predictions can be tested empirically, including how different crossing schemes may be used to identify signatures of intragenomic conflict.

## 2. Reproductive value, relatedness and intensity of kin competition modulate the potential for harm

Different portions of the genome flow between the sexes in different ways. These different patterns of transmission may consequently generate differences both in the reproductive values of males and females (i.e. the fraction of the ancestry that flows through them [36–38]), and in their relatedness to same-sex and opposite-sex patchmates. Such differences may therefore alter the value that males place upon their different social partners and thus modulate the evolutionary potential for male harm. To investigate the consequences of different modes of genetic transmission, we follow previous models of sexual conflict [24,26,28], considering a population subdivided into patches with the following life cycle: (1) $n_f$ adult female and $n_m$ adult males reside on each patch, (2) males compete to mate with the females on their patch, (3) females produce broods of offspring, (4) adult males and females die, (5) juveniles compete for breeding spots, with a proportion $a$ of the resulting competition occurring against natal patchmates, and (6) successful juveniles then become adults, starting the life cycle anew. A full description of the life cycle is given in the electronic supplementary material, figure S1.

Within this life cycle, we focus on a harming trait, expressed exclusively by males. This trait increases the relative competitiveness of its bearer (step 2), but decreases the fecundity of the females in his patch (step 3). Possible examples of such behaviour include male harassment, toxic ejaculates and mating plugs [4]. We determine the conditions under which natural selection favours an increase in the level of this harming trait, using the kin-selection approach of Taylor & Frank [39]. This approach analyses how the relative fitness of a focal individual is altered by both small changes in their own trait value and by correlated changes in the trait values of their social partners, with changes in relative fitness weighted by the reproductive value appropriate to their class [38] and the mode of inheritance exhibited by the focal locus [7,40]. See the electronic supplementary material, §1 for full details. This approach assumes weak selection and additivity, and as such it may be less informative for those alleles whose selective effects are particularly strong or highly non-additive. Applying

this methodology, we find that the condition for increase is given by

$$B[(c_{m \to f} + c_{m \to m})(1 - r_{mm})] - C[(1 - a_f)(c_{f \to f}\, r_{mf} + c_{m \to f}\, r_{mm}) + (1 - a_m)(c_{f \to m} r_{mf} + c_{m \to m} r_{mm})] > 0,$$

(2.1)

where $B$ is the scaled marginal benefit of increased competitiveness enjoyed by the focal individual male; $C$ is the scaled marginal cost of this harm upon the fecundity of the individual females with whom he interacts; $r_{ij}$ is the coefficient of genetic relatedness [5,7] between a sex-$i$ individual and a sex-$j$ individual drawn at random (with replacement) from the same patch (i.e. whole-group relatedness [41,42]); $c_{i \to j}$ is the class reproductive value [36–38] associated with gene-flow from sex-$i$ parents to sex-$j$ offspring; $a_i$ is the intensity of kin competition, i.e. the probability that sex-$i$ juvenile natal patchmates compete with one another for breeding spots (equivalent to the 'spatial scale of density-dependent competition' from [7]); and f and m indicate female and male, respectively.

Inspecting the left-hand side of condition (2.1), we can isolate the distinct selective effects of male harm, and the weightings placed upon them. The first portion captures the inclusive-fitness effect of increased mating success. This includes the direct benefit enjoyed by the focal male from increased mating success $B$, weighted by the reproductive value he accrues through his daughters $c_{m \to f}$ and sons $c_{m \to m}$, minus the concomitant loss of siring success by the average male on his patch (including himself) $-B$, weighted by his relatedness to them $r_{mm}$ and the reproductive value they would have accrued through their daughters $c_{m \to f}$ and sons $c_{m \to m}$. The second portion captures the inclusive-fitness effect of increased male harm upon female fecundity. This includes the loss of fecundity of female patchmates $-C$, weighted by the focal male's relatedness to these females $r_{mf}$ and the reproductive value they would have accrued through their daughters $c_{f \to f}$ and sons $c_{f \to m}$, and also the concomitant loss of fecundity of male patchmates who would have sired these lost offspring $-C$, weighted by the focal male's relatedness to these males $r_{mm}$ and the reproductive value they would have accrued through their daughters $c_{m \to f}$ and sons $c_{m \to m}$, with these losses of fecundity being defrayed to the extent $a_f$ and $a_m$ that competition for resources occurs among female and male natal patchmates, respectively.

We may rewrite condition (2.1) in the form $C/B < H$, where the dimensionless quantity $H$ defines the 'potential for male harm' [28,43] and is given by:

$$H = \frac{(c_{m \to f} + c_{m \to m})(1 - r_{mm})}{(1 - a_f)(c_{f \to f}\, r_{mf} + c_{m \to f}\, r_{mm}) + (1 - a_m)(c_{f \to m} r_{mf} + c_{m \to m} r_{mm})}.$$

(2.2)

The potential for harm summarizes the role of ecology, demography and transmission genetics in modulating the evolution of male harm, separate from the role of the more-contingent fecundity cost and benefit, providing a generalization of equation A6 in [28]. A larger potential for male harm means that harm is more likely to be favoured and, if favoured, is expected to be elaborated to a greater degree. Inspecting equation (2.2), we can see that: increasing relatedness (i.e. higher $r_{mf}$ and/or $r_{mm}$) will typically decrease the potential for male harm; increasing the intensity of kin competition (i.e. higher $a_f$ and/or $a_m$) will typically increase the potential for male harm, and increasing male reproductive value (i.e.

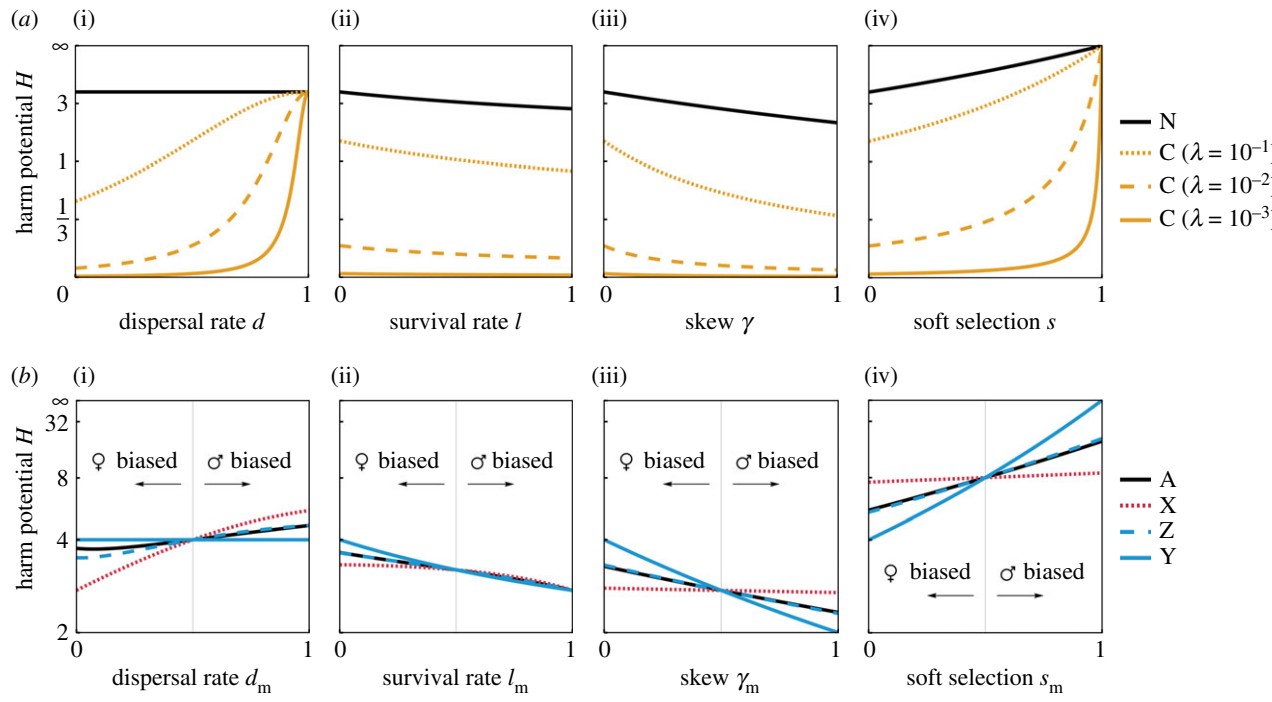

**Figure 1.** Demography modulates the potential for harm $H$ differently across the genome. (a) Demographic factors modulate harm differently between nuclear (N: autosomes, X, Y, Z) and cytoplasmic genes (C), with differences dependent on the extent of paternal transmission $\lambda$. In panels (ii–iv), $d = 0.5$. (b) Sex differences in demographic parameters such as (i) dispersal ($d_f = 0.5$), (ii) survival rate ($l_f = 0.5$), (iii) reproductive skew ($\gamma_f = 0.5$) and (iv) population regulation ($s_f = 0.5$), uncouple the interests of nuclear genes with respect to male harm. In panels (ii–iv) $d_f = d_m = 0.5$. Across all panels $n = 5$. Full methods to recreate these plots can be found in the electronic supplementary material, §2. (Online version in colour.)

higher $c_{m \to f}$ and/or $c_{m \to m}$) will typically increase the potential for male harm while increasing female reproductive value (i.e. higher $c_{f \to f}$ and/or $c_{f \to m}$) will typically decrease the potential for male harm.

Treating reproductive value, relatedness and kin competition as open parameters is useful for conceptualizing the higher-level forces shaping male harm, and generating comparative results that apply across a wide range of settings [7,34,35]. However, many specific ecological, demographic and genetic factors of interest will modulate several of these parameters simultaneously. For example, assumptions about which genetic party controls the trait, and thus the underlying transmission genetics, will shape both relatedness and reproductive value, and dispersal patterns will alter both relatedness and the intensity of kin competition. To understand how such concrete ecological, demographic and genetic factors will impact upon male harm, we now move from this 'open', more general model to a series of 'closed', ecologically, demographically and genetically explicit ones [34,35], in which the intensity of kin competition, relatedness coefficients and reproductive values emerge as functions of population processes, life cycle and transmission genetics. Specifically, we now investigate how dispersal, overlapping generations, reproductive skew and the mechanism of population regulation impact the potential for male harm across different parts of the genome.

## 3. Population viscosity drives conflict between nuclear and cytoplasmic genes

We first investigate how limited dispersal modulates the potential for male harm by considering that a fraction $1 - d$ of juvenile males and females remain on their natal patch, while a fraction $d$ disperse to other patches, prior to both

mating and reproduction (step 5). Lower dispersal increases relatedness between social partners (higher $r$), but also increases the intensity of competition between their offspring (higher $a$) [7,41]. For autosomes, as well as X, Y and Z chromosomes, we find that these two effects cancel exactly, such that the potential for male harm is completely independent of the dispersal rate and indeed is given by $H = n - 1$ across all of these genetic systems, where $n$ is the number of males on the patch (figure 1a(i)). That the potential for male harm is the same for autosomes and sex chromosomes under full dispersal ($d = 1$) recovers Andrés & Morrow's [44] result that there is no intragenomic conflict between these different portions of the genome in the absence of kin selection. Moreover, the invariance of the potential for male harm with respect to dispersal rate was shown previously for autosomes by Rankin [24] and Faria et al. [26]. Here we have shown that this invariance extends to the sex chromosomes such that, under the full range of dispersal rates ($0 \leq d \leq 1$), there is no intragenomic conflict with respect to male harm between the autosomes and sex chromosomes.

However, we find that this invariance does not extend to cytoplasmic elements (figure 1a(i); electronic supplementary material, §2.4). Assuming homoplasmy (i.e. that an individual's cytoplasmic factors are clonally related), and denoting the probability that a cytoplasmic gene is paternally inherited by $\lambda$ (analogous to the approach of [45]), then we find that this cancellation of increased local competition and increased relatedness only holds in the extremes of strict matrilineal ($\lambda = 0$), exact biparental ($\lambda = 0.5$) or strict patrilineal ($\lambda = 1$) inheritance. Outwith these three cases, the rate of dispersal modulates the potential for harm (figure 1a(ii)), and thus intragenomic conflicts can arise between cytoplasmic and nuclear genes. Under incompletely matrilineally biased inheritance ($0 < \lambda < 0.5$), lower dispersal is associated with

reduced potential for harm, and thus such cytoplasmic elements favour lower levels of harm than do nuclear ones. By contrast, under incompletely patrilineally biased inheritance $(0.5 < \lambda < 1)$, lower dispersal is associated with increased potential for harm, such that these elements favour a higher level of harm than do nuclear genes, although in this case the magnitude of the conflict is much smaller.

## 4. Further demographic factors shape relatedness and kin competition

Above we have shown that, for the various nuclear genes investigated (autosomes, X, Y and Z), limiting the rate of dispersal has no impact upon the potential for harm, owing to the way in which the effect of increased relatedness is perfectly offset by the effect of increased competition. In natural populations, other demographic factors will also typically be present in conjunction with limited dispersal, and together these may shift the balance between relatedness and kin competition, consequently modulating the potential for harm. To investigate this, we consider three further factors: overlapping generations, reproductive skew and soft selection (see electronic supplementary material, §2.1–4 for details).

Allowing adults to survive (and maintain their breeding spots) between generations with probability $l$ (i.e. at step 4 in the life cycle), we find that an increased rate of survival favours lower levels of harm (figure 1$a$(ii)). This occurs because higher survival increases relatedness between patchmates, but without altering the intensity of kin competition [46]. Consequently, the relatedness and competition effects of dispersal are decoupled in the context of overlapping generations, such that limiting the rate of dispersal leads to higher potential for male harm, but in a way that is exactly equal for autosomes and the various sex chromosomes. Similarly, we find that reproductive skew also decreases the potential for male harm (electronic supplementary material, table S2; figure 1$a$(iii)). If reproduction is skewed such that a few breeding adults contribute a disproportionate share of the offspring produced on the patch, then this will inflate the relatedness among patchmates while leaving the intensity of kin competition unchanged, thereby reducing the potential for harm. We integrate skew by defining a parameter $\gamma$, such that when $\gamma = 0$ all individuals enjoy the same fecundity under neutrality, while in the extreme of $\gamma = 1$, all juveniles share the same parents. We find that as the degree of skew increases, the potential for harm is reduced, and by exactly the same extent for autosomes and sex chromosomes. Decreasing patch size has a similar effect to increasing relatedness, reducing potential for harm (electronic supplementary material, table S2 and figure S2).

Finally, we consider the mechanism by which the population is kept constant in size, and the timing of this regulation step during the life cycle. In particular, we investigate the extent to which it occurs before versus after dispersal by allowing a proportion $s$ of regulation to occur before the dispersal phase and a proportion $1 - s$ of regulation after. Scenarios in which complete regulation occurs before dispersal, such that between-patch differences in productivity are completely abolished, have been described as involving 'soft selection' $(s = 1)$, whereas scenarios in which complete regulation occurs after dispersal, such that different patches may enjoy differences in productivity, have been described as involving 'hard selection' $(s = 0)$ [47–49]. Up to now, we have considered only hard selection $(s = 0)$. As we allow the proportion of regulation before dispersal $s$ to increase, so does the extent of kin competition $a$, and thus the potential for harm, with these effects equivalent for autosomes and sex chromosomes. In the limit of pure soft selection $(s = 1)$, decreased female fecundity does not alter the net productivity of the patch, and thus increased harm is always favoured $(H = \infty$; figure 1$a$(iv); electronic supplementary material, table S2). A fuller description of this life cycle, as well as other approaches to implement the effects of soft selection [50], can be seen in electronic supplementary material, §2.

## 5. Sex-biased demography drives intragenomic conflict between nuclear genes

Previous work has shown that, for autosomal genes, sex-biased demography may uncouple the balance between relatedness and kin competition [43,51] and consequently may modulate the potential for male harm [26]. Moreover, given their sex-specific inheritance patterns, these effects may be expected to be manifest differently across autosomes and sex chromosomes, thereby potentially uncoupling their inclusive-fitness interests and driving intragenomic conflicts of interest. To investigate this, we allow for sex biases in the rate of dispersal and survival, and in the degree of reproductive skew and soft selection.

We find that sex-biased dispersal $(d_f \neq d_m)$ leads to a divergence between the inclusive-fitness interests of autosomes and sex chromosomes (figure 1$b$(i)). Typically, under male-biased dispersal, the potential for male harm is greatest for X chromosomes and lowest for Y chromosomes $(H_X > H_Z > H_A > H_Y)$. Conversely, under female-biased dispersal this ranking is usually reversed $(H_X < H_Z < H_A < H_Y)$, and across these parameter values the Y chromosome remains invariant with respect to dispersal. However, this ranking of harm potential does not hold perfectly across all parameter values. For example, autosomes have the highest potential for male harm under high female-dispersal and low male-dispersal regimes. These complex patterns arise because sex-biased dispersal alters both the relatedness structure arising through matrilines and patrilines [27,52], and the intensity of competition experienced by daughters and sons. Owing to the sex-specific transmission patterns, these effects are felt differently by the different genomic elements. For instance, when male dispersal is low, kin competition is intense among sons relative to daughters, but patrilineal and matrilineal relatedness increase more evenly. This has a bigger harm-reducing effect for X chromosomes which are primarily transmitted through daughters (and thus experience a lesser increase in kin competition relative to the increase in relatedness).

Under sex-biased survival $(l_f \neq l_m)$, we find that sex chromosomes and autosomes once again diverge in their inclusive-fitness interests (figure 1$b$(ii)). With female-biased survival $(l_f > l_m)$, relatedness is higher through matrilines than patrilines; and with male-biased survival $(l_f < l_m)$, relatedness is higher through patrilines than matrilines [53]. Higher matrilineal relatedness has a greater impact upon those genomic elements for which a greater fraction are maternally inherited and thus the potential for harm is highest for Y chromosomes and lowest for X chromosomes $(H_Y > H_Z > H_A > H_X)$. The reverse is true when there is higher patrilineal relatedness, with harm lowest for Y chromosomes and highest for X chromosomes $(H_Y < H_Z < H_A < H_X)$. The same qualitative

pattern also obtains under sex differences in reproductive skew ($\gamma_f \neq \gamma_m$; figure 1$b$(iii)). When skew is higher in females ($\gamma_f > \gamma_m$) then there is higher matrilineal relatedness, and thus harm is highest for Y chromosomes and lowest for X chromosomes ($H_Y > H_Z > H_A > H_X$), and when skew is higher in males ($\gamma_f < \gamma_m$), then there is higher patrilineal relatedness, and thus harm is lowest for Y chromosomes and highest for X chromosomes ($H_Y < H_Z < H_A < H_X$). Similarly, sex biases in the number of breeders shapes relatedness. If there are more male than female breeders ($n_f < n_m$), then relatedness is higher through matrilines than patrilines, while if there are more female than male breeders ($n_f > n_m$), then relatedness is higher through patrilines than matrilines, with similar consequences as before on the potential for harm.

Finally, we find that the inclusive-fitness interests of autosomes and sex chromosomes also diverge as a consequence of sex-biased soft selection ($s_f \neq s_m$; figure 1$b$(iv)). If females experience a higher degree of soft selection, $s_f > s_m$, then kin competition is more intense among daughters than among sons. Conversely, if males experience a higher degree of soft selection, $s_f < s_m$, then the reverse obtains. Greater competition between same-sex relatives promotes harm more for those elements which achieve relatively higher reproductive value through that sex. Accordingly, when the degree of soft selection is greater in females then the potential for harm is lowest for Y chromosomes and highest for X chromosomes ($H_Y < H_Z < H_A < H_X$), and when the degree of soft selection is greater in males, then the potential for harm is highest for Y chromosomes and lowest for X chromosomes ($H_Y > H_Z > H_A > H_X$). Alongside these potentials for harm, we also analyse an example with specific male and female fecundity functions in electronic supplementary material, §4, explicitly solving for the optimum harm value across different loci (electronic supplementary material, figures S3–S5).

## 6. Discussion

Male harming traits have been described across a wide range of taxa [4], from traumatic insemination of bed bugs [54], and grasping appendages of water striders [55,56], to proteins in the ejaculates of flatworms [57], and tomiodonts of painted turtles [58]. Recent theory has shown how kin selection may curb the worst excesses of such male harm [24–27,29] and has been supported empirically in a growing range of taxa, including arachnids, birds and insects [12–21,23,30]. We have built upon this theory to show how aspects of demography may shape the potential for male harm differently across different parts of the genome, yielding novel predictions as to how intragenomic conflicts may emerge over such traits, where male harm loci are likely to be enriched, and how these patterns are expected to vary across different populations and species.

In particular, we have found that cytoplasmic genes may favour distinct levels of harm to their nuclear counterparts. As matrilineal inheritance of both cytoplasmic genes and other endosymbionts is the norm across the animal and plant kingdoms [59], our analysis suggests that these elements tend to favour lower levels of male harm than do nuclear genes, generating potentially intense intragenomic conflicts over such traits. One particular arena of conflict may be over sperm competitiveness, as while competitive sperm may provide a benefit to the focal male, they may also impose fecundity costs for

females, for example through zygote inviability owing to polyspermy [60,61]. Given the central role mitochondria play in sperm physiology, there may be a large scope for conflict in this context. Indeed, while mitochondrial alleles contributing to variation in sperm performance are typically assumed to be the products of drift [62–64] (i.e. 'mother's curse' [65,66]), in viscous populations such alleles may be positively selected if they reduce the fitness costs imposed upon interacting female kin (this is a negative variant of the argument made by [67]). It also mirrors the evolution of male-killing symbionts in various arthropod groups [68]; in both cases, alleles which decrease a male's fitness may improve the fitness of his female relatives, by either reducing the extent of male harm or decreasing the intensity of juvenile competition for resources.

Although matrilineal inheritance is the norm for cytoplasmic elements, various exceptions—such as the doubly uniparental inheritance of bivalve molluscs [69], paternal transmission of mitochondria in cucumbers and sequoias [70] and paternal transmission of symbionts in mosquitos [71], leafhoppers [72] and tsetse flies [73]—provide exciting avenues for further empirical testing, with non-matrilineally inherited genes expected to exhibit greater harm than those with strict matrilineal inheritance. Although the above examples are somewhat speculative, one example which may be more amenable to experimental investigation is the obligate vertically transmitted rhabdovirus sigma. This is biparentally transmitted in *Drosophila melanogaster* [74], and experiments have shown that males infected with sigma appear to have increased mating success [75], although the mechanism of action and direct cost to females (if any) is unclear. Given that sigma viruses infect other arthropods and appear to show similar transmission patterns [76], this system may be amenable for comparisons across different transmission patterns and demographic scenarios, as well as to experimental manipulation of these factors. Moreover, although we have made a conceptual distinction between nuclear versus cytoplasmic genes, there are nonetheless nuclear genes whose inheritance patterns more closely match those of cytoplasmic factors, and to which our predictions for cytoplasmic genes may readily apply. For instance, the germline-restricted chromosome in zebra finches is maternally transmitted, with rare occurrences of paternal transmission [77,78]. Depending on which tissues such genes are present in, and the extent of their expression, then these too may have the potential to modulate male harm and come into conflict with other genes inhabiting the same nuclei.

We have also shown that while the potential for harm is constant across the nuclear genes under sex-neutral demography, population viscosity in concert with sex-biased demography generates differences in how male harm loci evolve on autosomes and sex chromosomes. This shares conceptual similarities with how the potential for altruistic behaviour remains invariant across diploidy and haplodiploidy under sex-neutral dispersal but diverges under sex-biased dispersal [41,51,79]. This yields predictions about where male harm loci should be enriched across the genome, and how such patterns will depend on both the extent and direction of sex biases in demography. Currently, these predictions are challenging to test, as the genetics of many male harm traits is still poorly understood [32]. As Rowe *et al.* point out [32], there are cases where the phenotypic interactions are well understood, but the genetics is not, and cases where the genetics is well understood, but the phenotypic interactions are not. Nonetheless, there are an increasing number of examples that span this

gap, including genes underpinning the morphological aspects of grasping behaviours in water striders [80,81], metabolic genes associated with siring success in bulb mites [82–84], and gamete-recognition proteins in abalone and sea urchins [85,86]. Alongside these specific examples, there are classes of male harm traits that may be particularly amenable to large-scale genetic analysis. For example, sperm fluid proteins (Sfps) currently represent among the most-successful syntheses of the genetics and ecology of sexual conflict [87]. Though even here we lack detailed knowledge of what many of these proteins do, and whether or not they are definitively involved in sexual conflict. For instance, while the *Drosophila melanogaster* seminal proteome is well-characterized in comparison with many others, we still only have a good functional understanding of about 10% of its constituent proteins [88]. As the proteomes of more species have begun to be characterized, and those proteomes functionally described, then hopefully these within-genome comparisons will be increasingly tractable.

In addition to studies of natural populations, experimental evolution offers ways to artificially generate particular population structures and thereby investigate their effects on male harm. Previously, such approaches have been used in bulb mites [17], spider mites [18] and seed beetles [20,23]. While spider mites are arrhenotokous and lack sex chromosomes, both bulb mites (XO) and seed beetles (XY) have sex chromosomes and therefore may be systems in which the predictions we have outlined could be most effectively tested. Additionally, as well as manipulating population structures, previous studies in *D. melanogaster* have used balancer chromosomes to manipulate the inheritance patterns experienced by X chromosomes, enforcing either strictly matrilineal or strictly patrilineal inheritance (e.g. [89,90]). Accordingly, combined with manipulated populations structures, one could effectively force Y-chromosomal inheritance on the X, and thus partially control for the historical gene content of these different chromosomes, as well as enabling experimental exploration of the intervening parameter space.

Even if individual trait levels are dominated by the interests of autosomal genes—as might be expected if they are the largest genomic faction [91–93]—intragenomic conflicts may still be expected to escalate and lead to differences in the abundance of 'harm-promoting' and 'harm-inhibiting' loci across different portions of the genome [27,94]. Population crosses can provide one method to identify these signatures of intragenomic conflict, by creating 'imbalanced' genomes with a relative abundance or paucity of harm-promoting or harm-inhibiting loci (see electronic supplementary material, §4) [27]. For instance, evidence of conflicts between maternal-origin and paternal-origin genes have been found by performing reciprocal crosses in flowering plants [95–99], mammals [100–102] and insects [103–105]. Similar approaches may be used to uncover the intragenomic conflicts we have outlined here. In electronic supplementary material, figure S6, we present two examples of the phenotypes predicted for a cross between two populations with an XO sex-determination system. If these two populations differ in either the direction or intensity of conflict, then crosses between them are expected to lead to extreme phenotypes as the delicate balance between competing genomic factions is disrupted, with reciprocal directions of cross expected to lead to opposite phenotypes. Given the extreme phenotypes that are expected to arise from population crosses, this may be a further mechanism by which intragenomic conflict contributes to hybrid inviability and hence speciation [106].

Our analysis has focused on cases where males reduce the immediate fecundity of interacting females, but population structure is likely to uncouple the interests of different genomic factions in relation to a wider set of both inter- and intra-sexual social traits. This may include other forms of male harm, for example whereby harm reduces longevity and/or future fecundity or reduces fecundity unevenly across the females in the group [28,107]. While some of these effects are already incorporated in our model—for instance, when there is no survival of females between generations ($l_f = 0$), a reduction in female reproductive success could be interpreted as owing to loss of fecundity or alternatively premature death before the completion of reproductive effort—more generally these different assumptions about the ecology of harm will probably alter results, at least quantitively if not qualitatively. Moreover, we have only investigated the selective pressures shaping male harm, and thus have not considered possible coevolutionary dynamics between male harm and female resistance. Previous analyses of these coevolutionary processes have typically focused on autosomal inheritance (e.g. [24,26,27,108,109]) but the inclusion of other genomic elements may well lead to divergent results, because—as we have shown—these elements may have distinct evolutionary interests. Coevolutionary dynamics will therefore probably be sensitive to assumptions about demography, but also to how control over the phenotype is dispersed across the genome and the relative phenotypic power these genomic elements have in males and females. Recent theoretical and empirical work in *D. melanogaster* has shown that sexually antagonistic coevolution of the sex chromosomes may also play an important role in speciation [31], and thus we may expect the intragenomic conflicts that we have outlined here to further contribute to the origin of species [106].

To conclude, we have shown how ecological and demographic processes—and, in particular, their sex-specific aspects—may differentially mould male harm across the various inheritance systems that coexist within individual genomes. With differences in the flow of genes from mothers and fathers to daughters and sons, differences in relatedness to social partners and the intensity of kin competition may emerge, igniting conflicts of interest between autosomes, sex chromosomes and cytoplasmic elements over male harm. As knowledge of the molecular basis of sexual conflict grows—from flies and water striders to abalone and sea urchins—these models may help guide the design of future experiments and aid in the interpretation of data collected from natural populations.

Data accessibility. The data are provided in the electronic supplementary material [110].

Authors' contributions. T.J.H.: conceptualization, formal analysis, investigation, methodology, project administration, validation, visualization, writing—original draft, writing—review and editing; A.G.: conceptualization, funding acquisition, methodology, resources, supervision, writing—review and editing.

All authors gave final approval for publication and agreed to be held accountable for the work performed therein.

Competing interests. The authors declare no conflicts of interest.

Funding. T.J.H. is supported by a PhD scholarship funded by the School of Biology, University of St Andrews. A.G. is supported by a Natural Environment Research Council Independent Research Fellowship (grant no. NE/K009524/1) and a European Research Council Consolidator (grant no. 771387).

Acknowledgements. We thank G. Faria, V. Litzke, J. Rayner, M. Ritchie, D. Shuker and K. Stucky for helpful discussion, and two anonymous reviewers for constructive comments.

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
