## [Peer Review File · Proceedings of the Royal Society B: Biological Sciences]

Review History

RSPB-2021-1461.R0 (Original submission)

Review form: Reviewer 1

Recommendation

Accept with minor revision (please list in comments)

Scientific importance: Is the manuscript an original and important contribution to its field?

Good

General interest: Is the paper of sufficient general interest?

Good

Quality of the paper: Is the overall quality of the paper suitable?

Excellent

Is the length of the paper justified?

Yes

Should the paper be seen by a specialist statistical reviewer?

No

Do you have any concerns about statistical analyses in this paper? If so, please specify them explicitly in your report.

No

It is a condition of publication that authors make their supporting data, code and materials available - either as supplementary material or hosted in an external repository. Please rate, if applicable, the supporting data on the following criteria.

Is it accessible?

Yes

Is it clear?

Yes

Is it adequate?

Yes

Do you have any ethical concerns with this paper?

No

Comments to the Author

This paper presents an interesting gap in the literature and addressed it in a theoretical manor which seemed very suitable. They investigate how different modes of inheritance (via autosomes, sex chromosomes, or cytoplasmic elements) have potentially different evolutionary outcomes due to differences in demography. They approach the problem first with a non-explicit mathematical model and then expand to an explicit version of the model, allowing them to investigate more questions. I thought the paper was written well and I appreciated the thorough approach to the relevant literature – well done!

Whilst I thought the paper was well written, I have some points that I think need to be addressed to help make the manuscript as transparent and clear as possible. These points are as follows –

- Virtually all the mathematics was stored in the supplementary material. I do agree that it makes sense to store a lot of it in there, as there are plenty of equations in there which may be daunting to a casual reader. However, I think might be helpful for the audience if you could add some discussion about how you are deriving these conditions for increasing male harm in a general sense. For example, describing the kin-selection approach of Taylor and Frank you mention on line 77. I think this might make the results clearer without making the reader have to go into the supplementary information to understand how the method works.
- As I have understood it, the authors have only considered male harm in the context of affecting the fecundity of the female he mates with. This seems to ignore the potential for a mortality effect on females due to male harm, which is likely. I think there should be some discussion of how such a mortality effect might affect their results even if the authors do not investigate this as thoroughly as the fecundity effect.
- It was not completely clear to me what the authors meant by population regulation. For example, is that density-dependent mortality, a generational mortality, or some other mechanism. I was unsure exactly what the authors meant and because of that its harder to understand the significance of the differences between the DRD and RDR selection systems. It would be helpful if the authors could clarify exactly what they mean and the subsequent significance for the different selection regimes.
- The authors present a lot of results which is great, but it does make finding the key results and other results one may be interested in harder to identify. I think it would be helpful if the authors could add something like a table to summarise all the results.
- I personally think the mathematical approach used is suitable and makes sense, but my viewpoint is that of a mathematician. In the interest of a more general reader, I think it could be helpful to briefly mention some of the strengths and weaknesses of the approach used.

I have a few more comments about the supplementary material that I think are just as important to address, since that is where the technical detail is presented. In general, the points can be summarised as wanting more justification and description of the function choice, but I will outline a couple of specific cases below to try and make my point clear.

- In the description of Table S1, $a_{\{f\}}$ and $a_{\{m\}}$ are defined for both the DRD and RDR selection models. I would like to see some explanations as to why they take these forms and why these forms are biologically suitable definitions.
- Likewise, equations (S9a-f), (S11a-f) give definitions for the relative fitness, however it does not seem clear why they are of this form. Have you derived these from first principles, chosen them arbitrarily, or followed others work? Whichever it is, it would be nice to highlight where these equations have come from, and why they are suitable functional forms.

Review form: Reviewer 2

Recommendation

Major revision is needed (please make suggestions in comments)

Scientific importance: Is the manuscript an original and important contribution to its field?

Good

General interest: Is the paper of sufficient general interest?

Good

Quality of the paper: Is the overall quality of the paper suitable?

Acceptable

Is the length of the paper justified?

Yes

Should the paper be seen by a specialist statistical reviewer?

No

Do you have any concerns about statistical analyses in this paper? If so, please specify them explicitly in your report.

No

It is a condition of publication that authors make their supporting data, code and materials available - either as supplementary material or hosted in an external repository. Please rate, if applicable, the supporting data on the following criteria.

Is it accessible?

N/A

Is it clear?

N/A

Is it adequate?

N/A

Do you have any ethical concerns with this paper?

No

Comments to the Author

Summary

In this paper, the authors investigate the evolution of male harm (a form of interlocus sexual conflict where males increase their siring success at the expense of females' fecundity) across different genomic locations in group structured populations such that individuals interact and compete among relatives. They do so by analysing the conditions that favour the invasion of an allele at a locus coding for male harm when this locus can sit on an autosome, a X-, Z-, or Y-chromosome, or a cytoplasmic genomic element. A number of demographic parameters that are relevant to group-structured populations are also allowed to vary (including in a sex-specific manner), such as dispersal, adult survival, degree of reproductive skew, and whether regulation occurs before or after dispersal. They find that the different nuclear genomic locations have exactly the same propensity to favour male harm when demographic parameters are equal across males and females (e.g. male and female juveniles disperse at the same rate). By contrast, the potential for cytoplasmic genomic elements to harbour male harm alleles can be sensitive to demography (provided inheritance is not strictly uniparental). Where nuclear genomic locations do differ in their potential for male harm is where demographic parameters differ among the sexes, such as dispersal rate or timing. They use their result to discuss the genomic distribution of male harm as well as potential intra-genomic conflict for male harm.

Overall, we have a positive impression of this manuscript. The introduction and discussion are well-written and should eventually be useful for future discussions on the genomic location of alleles influencing male harm. The breadth of scenarios investigated, in particular, is commendable.

There are however also a number of major issues with the current version of this manuscript. Two seem particularly important. First, there is a significant lack of clarity about the assumptions underlying some of the models. This makes following the authors' arguments challenging at times, even for theoretical biologists, and ultimately clouds the biological interpretation of the results. Second, and probably more troublesome from an empirical perspective, is that many of the reported effect sizes seem small. This in itself is not a problem (at least to us) but it sits at odds with the language employed in the main text, with as a consequence the main claims of the paper looking a bit overplayed. We expand on these points below.

Major Points

The assumptions behind the "open model" are too ambiguous. The main text suggests that eq. (1) applies to a broad set of models whose assumptions are those mentioned on L68-71. First it is not clear to us what is meant by "with a proportion of the resulting competition occurring locally, in a sex-specific manner". What is a proportion of competition exactly? It is the ratio of which quantities? In particular, how is it connected to individual processes of competition for resources, competition for breeding spots and dispersal?

Second, are L68-71 really the only assumptions required for eq. (1)? Shouldn't there at least also be random mating (and interactions among the sexes) within patches? This should be mentioned given that the paper is about the evolution of such interactions. Shouldn't dispersal be uniform and random among patches? Shouldn't also the number of adult males and females be fixed and constant in all patches (otherwise the selection gradient i.e. eq. (1), should incorporate the effects of traits on local demography, see Rousset & Ronce, 2004 for inclusive fitness for traits affecting metapopulation demography). Actually, don't equations S1 in the appendix suggest that the sex ratio among adults is 1:1 ($n_m = n_f$) as equations S1 tell us that in the absence of genetic variation (when $x=y=z$) a male produces on average one female and one female produces one male? This is a non-exhaustive list. As is expected from a theory paper, the authors should be clear about what types of traits and populations can be understood with eq. (1).

Another confusing aspect of eq. (1) is how quantities are verbally defined. From the introduction and the rest of the manuscript, we expect all quantities to be defined at the genetic level in eq. (1) as eq. (1) is supposed to be able to consider alleles that can sit on different chromosomes with different patterns of segregation and sex-specificities. In other words, we expect the fitness effects (B and C), the relatedness coefficients (r), reproductive values (c) and competition effects to be defined in terms of a focal gene rather than a focal individual. How do you navigate between the different genetic models from eq. (1) e.g. how do you go from an autosome to a Y-chromosome? Is that all contained in the class reproductive values? If so, statements like “the reproductive value they would have accrued through their daughters $m \rightarrow f$ and sons $m \rightarrow m$ ” (L98), where “they” refers to a focal male, are misleading. “They” should refer to gene copies.

About relatedness coefficients: L83 informs us that they look at the degree of relatedness between two individuals sampled at random with replacement (i.e. including the focal). Then it is not clear to us how the authors arrive to the following interpretation for the first term of eq. (1) on L89-91: “the direct benefit enjoyed by the focal male from increased mating success, [...] minus the mating success lost by his mate competitors —, weighted by his relatedness to them r_{mm} ” (where according to L83 r_{mm} is calculated with replacement). But shouldn't the relatedness between a focal male and its competitors (i.e. the other males in the patch) be calculated “without replacement”? Connected to this, it would be helpful if the authors were clearer about when (and how) the competition among males for siring, and male harm to females occur.

It is probably clear by now that at least to us, the connection between eq. (1) and the traits, their genetic basis, the life cycle and their fitness effects is difficult to grasp, even from reading the appendix. Given its current ambiguity and the seemingly false sense of generality that it gives, we would advise to drop the open model as such. The decomposition of the selection gradient according to the kin selection approach is of course extremely useful to interpret results. In our opinion, it would therefore be more powerful and easier to follow if the selection gradient was presented for a particular “closed model”, which can be easily tied to clear biological processes, but in such a way that it is decomposed according to eq. (1) with all the relevant effects of interactions and competition among relatives to explain selection and models to come.

The closed models also suffer from a certain lack of clarity, in particular the soft selection cases where $0 < s < 1$. Firstly, it is unclear what the biology of the models represents, and therefore how they modulate the soft/hardness of selection. In particular it is difficult to see what the parameter s actually represents if under DRD it refers to “the proportion of regulation occurring before dispersal” while under RDR it refers to the “proportion of dispersal occurring after regulation”? The ambiguity here is accentuated by the fact that appendix eqs. S9-12 lack any explanatory text making them opaque. Put briefly, it is difficult to understand what is supposed to be happening at the individual level in these models (e.g. how would one program this life-cycle in an individual-based simulation). One way to address these issues would be to clarify how the parameters in these models relate to those used in previous studies. For example, what is the relationship between the life-cycle for the DRD case and the model in Debarre & Gandon 2011 (see their figure 2)? Specifically, in Debarre & Gandon there are two dispersal parameters and (d_j, d_A) and an additional parameter q , but it's not clear how these parameters relate to d_m, d_f, s_m, s_f in the present paper.

Furthermore, because the two soft selection models are hard to understand, it's difficult to see how they're different from each other and therefore what the reason behind modelling both cases is. Similarly, no explanation is given as to why there are differences in harm potential between the two cases when s is intermediate as mentioned on L192-3. Additionally, the description of results given on L193-7 is also confusing. Firstly, it is stated that soft selection leads to reduction in harm potential on L193-4 but figures 2c-d appear to show the opposite pattern. Secondly, on L194-5 the authors say that the invariance of harm potential with respect to dispersal is maintained under RDR but not DRD, but it is not clear where we are supposed to see this result as figures 2c-d and 3c-d both show output for a fixed level of dispersal.

Overall, much of the lack of clarity in this paper owes to the Appendix being frustratingly cursory, making rederivation and interpretation of the results near impossible. Many important equations are presented with minimal or no explanatory text (e.g. S9-12) and so are difficult to follow intuitively. To allow readers to follow the arguments behind the model, it is essential to provide more text that explain derivations, especially of the fitness equations.

Another issue for the paper's reproducibility is the weakness of the links between the Appendix and main text. For example the majority of the paper's results are presented numerically as figures, yet there is no explanation in either the main text figure legends or the Appendix as to how the figures are generated (e.g. which equations are used and what they are solved for in order to arrive at each of the curves). Each section of main text results should reference specific locations in the appendix where results are derived, and vice-versa, e.g. presumably the results in the section "Population viscosity drives intragenomic conflict between nuclear and cytoplasmic genes" are generated by analysing the closed model appendix equations and setting s to 0 and $dm = df$, but this needs to be clearly stated somewhere. In other words, we need to be able to recreate the paper's figures through a few guided substitutions, which is not the case currently.

From a more empirical perspective, an important question is what are the units of "harm potential" (H) in the context of the amount of harm that is apportioned to females? For example, what are the consequences for female fecundity of a 1 unit increase in H ? To help, a more specific explanation of H after eq. 2 is needed - it appears that H gives the number of offspring of a random female in its patch that a male will sacrifice through harm in order to increase his own direct reproduction by one offspring, is this accurate? The ambiguity over how H is scaled also makes it difficult for the reader to interpret the effect sizes shown in the figures. In fact, a number of the reported effect sizes appear relatively small upon inspection. Specifically, the magnitude of the differences between nuclear genes in response to demography appears to be quite small in Figures 1b and especially 3a-b (which are scaled between 0.65 and 0.85) when plotted as $H/(1+H)$. This is not consistent with the relevant sections of the main text results. In addition, the potential for effect sizes makes it difficult to interpret the notion of hypo- and hyper- harming presented in the discussion and fig S1. To what degree would H increase/decrease in these cases and quantitatively what would the effects be for female fitness?

The issue over ambiguous effect sizes also draws into question the tone of the paper. Currently, the manuscript's abstract, discussion (and title!) rely on the demonstration that demography modulates conflict across genomic regions. However such a conclusion is not especially straightforward if many of the effect sizes are in fact small. In order to avoid overstating, we therefore suggest some changes to discussion and abstract are warranted, or clarification of the results to more convincingly show significant differences among genomic regions due to sex-specific demography.

In a similar vein, it is also clear from the analyses that selection on male harm is invariant across different nuclear regions for a number of common demographic cases (i.e. sex-congruent dispersal, reproductive skew, degree of soft selection). The consistency of this result across many genomic regions is not uninteresting and there should be a paragraph discussing the causes/implications of this in the discussion.

Minor Points

Main text:

L84 - should it be "gene flow"? And "i" and "j" be italicised here?

L84 - define the scale of competition.

L109 - unclear where "increasing the scale" means; as increases, more individuals compete locally, is this really an increase in scale?

L125 - state this is sex-independent dispersal.

L146 - reference a figure, in general the results section would benefit from more frequent and specific references to figures (e.g. which colour curves the reader should be looking at)

L166-7 - cite Taylor & Irwin, 2000 here.

L192 - word missing.

L207 - Could you please explain your result that “Typically, with male-biased dispersal, the potential for male harm is greatest for X chromosomes, and lowest for Y chromosomes ($> > >$). Conversely, when there is female-biased dispersal this ranking is usually reversed ($< < <$).” at greater length? Since females never carry a Y, then males never harm genetic relatives at the harming locus. However, they compete with genetic relatives at the harming locus when dispersal is limited, especially in males. So one could expect that the potential for harm is greater when dispersal is female biased.

L238 - “there” typo.

L274 - Could you be more specific about how you would test for the effects of non uniparental inheritance on male harm in these species? For instance what type of male harm has been observed in those species listed?

L366 - This “framework will guide” → This “model may help” perhaps?

Figure 3 Legend - What is dm?

Appendix:

L12 - Could you define absolute and relative fitness precisely? E.g. is $W_f \rightarrow f$ the expected total number of female offspring produced by a female offspring (including in philo- and parapathy)?

L21 - Could you give a few examples of how the breeding value would change according to looking at different genomic locations?

L33A - It’s confusing to use here if it also refers to reproductive skew later on.

L51A - space missing.

L48-57 - Make it clearer here what “survival” means (i.e. between-generations), as previously in the manuscript survival is also used in reference to reaching the adult mating pool as described in L3-4.

L59-60 - space missing (after DRD and RDR).

Eq. S2 - Please define W clearly.

Eq. S9 and S11: These need to be clearly explained.

Eq. S10 and S12: How are these equations connected to eq.S1? The open model was phrased as if it could be applied to any model but it seems difficult to express Eqs. S10 and S12 as eqs.S1.

L93 - Perhaps explain why alpha and beta are useful and how they allow you to consider different genetic loci.

Eq. S24 - Why are these probabilities useful ? i.e. when do you use them ?

L.147 - This section is much too terse. We need to know more clearly how we can reproduce your results.

Refs

Rousset, F. and Ronce, O., 2004. Inclusive fitness for traits affecting metapopulation demography. *Theoretical population biology*, 65(2), pp.127-141.

Taylor, P. D. 1992a. Altruism in viscous populations: an inclusive fitness model. *Evol. Ecol.* 6:352–356.

Taylor, P. D. 1992b. Inclusive fitness in a homogeneous environment. *Proc. R. Soc. Lond. B Biol. Sci.* 249:299–302.

Taylor, P. D., & Irwin, A. J. 2000. Overlapping generations can promote altruistic behavior. *Evolution*, 54(4), 1135-1141.

Wilson, D. S., G. B. Pollock, and L. A. Dugatkin. 1992. Can altruism evolve in a purely viscous population? *Evol. Ecol.* 6:331–341.

Decision letter (RSPB-2021-1461.R0)

02-Aug-2021

Dear Mr Hitchcock,

We have now received referees' reports on your manuscript RSPB-2021-1461 entitled "Sex biased demography modulates male harm across the genome".

The manuscript has, in its current form, been rejected for publication in Proceedings B. This action has been taken on the advice of the Associate Editor and the referees, who have recommended that substantial revisions are necessary. With this in mind we would be happy to consider a resubmission, provided the comments of the referees are fully addressed. However please note that this is not a provisional acceptance.

Yours sincerely,
Professor Loeske Kruuk
Editor
<mailto:proceedingsb@royalsociety.org>

Associate Editor
Board Member: 1
Comments to Author:

The manuscript was reviewed by two experts and myself. We all agreed that this is a high-quality paper that addresses an important gap in the literature. Here, the authors investigate how demography shapes the evolution of male harm traits, with a particular focus on how mode of inheritance influences the outcome. The topic is of broad interest to a range of topics, including evolution, sexual selection, kin selection, demography and genome evolution.

However, both reviewers raised a number of questions that need to be addressed.

Most importantly:

- 1) Both reviewers commented on the lack of clarity regarding the mathematics and assumptions of the model. It is clear that additional explanation and justification is required.
- 2) Both reviewers also commented on the disjunction between the supplementary material (where the mathematics of the model is outlined) and the main body of the manuscript. The paper would benefit from clearer signposting and a better explanation of the conditions of the model. Importantly, this should be understandable to non-theoreticians.
- 3) Related to the above points, the paper would benefit from a clearer explanation of the main predictions and results of the model. The first reviewer suggests including a table to summarise the main results. This may also help to deal with the second reviewer's concerns about effect sizes. This is an important point that should be addressed fully.
- 4) Finally, the second reviewer raises a number of thoughtful questions about specific assumptions and parameters of the model that should be addressed.

Reviewer(s)' Comments to Author:

Referee: 1

Comments to the Author(s)

This paper presents an interesting gap in the literature and addressed it in a theoretical manor which seemed very suitable. They investigate how different modes of inheritance (via autosomes, sex chromosomes, or cytoplasmic elements) have potentially different evolutionary outcomes due to differences in demography. They approach the problem first with a non-explicit mathematical model and then expand to an explicit version of the model, allowing them to investigate more questions. I thought the paper was written well and I appreciated the thorough approach to the relevant literature – well done!

Whilst I thought the paper was well written, I have some points that I think need to be addressed to help make the manuscript as transparent and clear as possible. These points are as follows –

- Virtually all the mathematics was stored in the supplementary material. I do agree that it makes sense to store a lot of it in there, as there are plenty of equations in there which may be daunting to a casual reader. However, I think might be helpful for the audience if you could add some discussion about how you are deriving these conditions for increasing male harm in a general sense. For example, describing the kin-selection approach of Taylor and Frank you mention on line 77. I think this might make the results clearer without making the reader have to go into the supplementary information to understand how the method works.
- As I have understood it, the authors have only considered male harm in the context of affecting the fecundity of the female he mates with. This seems to ignore the potential for a mortality effect on females due to male harm, which is likely. I think there should be some discussion of how such a mortality effect might affect their results even if the authors do not investigate this as thoroughly as the fecundity effect.
- It was not completely clear to me what the authors meant by population regulation. For example, is that density-dependent mortality, a generational mortality, or some other mechanism. I was unsure exactly what the authors meant and because of that its harder to understand the significance of the differences between the DRD and RDR selection systems. It would be helpful if the authors could clarify exactly what they mean and the subsequent significance for the different selection regimes.
- The authors present a lot of results which is great, but it does make finding the key results and other results one may be interested in harder to identify. I think it would be helpful if the authors could add something like a table to summarise all the results.
- I personally think the mathematical approach used is suitable and makes sense, but my viewpoint is that of a mathematician. In the interest of a more general reader, I think it could be helpful to briefly mention some of the strengths and weaknesses of the approach used.

I have a few more comments about the supplementary material that I think are just as important to address, since that is where the technical detail is presented. In general, the points can be summarised as wanting more justification and description of the function choice, but I will outline a couple of specific cases below to try and make my point clear.

- In the description of Table S1, $a_{\{f\}}$ and $a_{\{m\}}$ are defined for both the DRD and RDR selection models. I would like to see some explanations as to why they take these forms and why these forms are biologically suitable definitions.
- Likewise, equations (S9a-f), (S11a-f) give definitions for the relative fitness, however it does not seem clear why they are of this form. Have you derived these from first principles, chosen them arbitrarily, or followed others work? Whichever it is, it would be nice to highlight where these equations have come from, and why they are suitable functional forms.

Referee: 2

Comments to the Author(s)

Summary

In this paper, the authors investigate the evolution of male harm (a form of interlocus sexual conflict where males increase their siring success at the expense of females' fecundity) across different genomic locations in group structured populations such that individuals interact and compete among relatives. They do so by analysing the conditions that favour the invasion of an allele at a locus coding for male harm when this locus can sit on an autosome, a X-, Z-, or Y-chromosome, or a cytoplasmic genomic element. A number of demographic parameters that are relevant to group-structured populations are also allowed to vary (including in a sex-specific manner), such as dispersal, adult survival, degree of reproductive skew, and whether regulation occurs before or after dispersal. They find that the different nuclear genomic locations have exactly the same propensity to favour male harm when demographic parameters are equal across males and females (e.g. male and female juveniles disperse at the same rate). By contrast, the potential for cytoplasmic genomic elements to harbour male harm alleles can be sensitive to demography (provided inheritance is not strictly uniparental). Where nuclear genomic locations do differ in their potential for male harm is where demographic parameters differ among the sexes, such as dispersal rate or timing. They use their result to discuss the genomic distribution of male harm as well as potential intra-genomic conflict for male harm.

Overall, we have a positive impression of this manuscript. The introduction and discussion are well-written and should eventually be useful for future discussions on the genomic location of alleles influencing male harm. The breadth of scenarios investigated, in particular, is commendable.

There are however also a number of major issues with the current version of this manuscript. Two seem particularly important. First, there is a significant lack of clarity about the assumptions underlying some of the models. This makes following the authors' arguments challenging at times, even for theoretical biologists, and ultimately clouds the biological interpretation of the results. Second, and probably more troublesome from an empirical perspective, is that many of the reported effect sizes seem small. This in itself is not a problem (at least to us) but it sits at odds with the language employed in the main text, with as a consequence the main claims of the paper looking a bit overplayed. We expand on these points below.

Major Points

The assumptions behind the "open model" are too ambiguous. The main text suggests that eq. (1) applies to a broad set of models whose assumptions are those mentioned on L68-71. First it is not clear to us what is meant by "with a proportion of the resulting competition occurring locally, in a sex-specific manner". What is a proportion of competition exactly? It is the ratio of which

quantities? In particular, how is it connected to individual processes of competition for resources, competition for breeding spots and dispersal?

Second, are L68-71 really the only assumptions required for eq. (1)? Shouldn't there at least also be random mating (and interactions among the sexes) within patches? This should be mentioned given that the paper is about the evolution of such interactions. Shouldn't dispersal be uniform and random among patches? Shouldn't also the number of adult males and females be fixed and constant in all patches (otherwise the selection gradient i.e. eq. (1), should incorporate the effects of traits on local demography, see Rousset & Ronce, 2004 for inclusive fitness for traits affecting metapopulation demography). Actually, don't equations S1 in the appendix suggest that the sex ratio among adults is 1:1 ($n_m = n_f$) as equations S1 tell us that in the absence of genetic variation (when $x=y=z$) a male produces on average one female and one female produces one male? This is a non-exhaustive list. As is expected from a theory paper, the authors should be clear about what types of traits and populations can be understood with eq. (1).

Another confusing aspect of eq. (1) is how quantities are verbally defined. From the introduction and the rest of the manuscript, we expect all quantities to be defined at the genetic level in eq. (1) as eq. (1) is supposed to be able to consider alleles that can sit on different chromosomes with different patterns of segregation and sex-specificities. In other words, we expect the fitness effects (B and C), the relatedness coefficients (r), reproductive values (c) and competition effects to be defined in terms of a focal gene rather than a focal individual. How do you navigate between the different genetic models from eq. (1) e.g. how do you go from an autosome to a Y-chromosome? Is that all contained in the class reproductive values? If so, statements like "the reproductive value they would have accrued through their daughters $\frac{c}{2}m \rightarrow f$ and sons $\frac{c}{2}m \rightarrow m$ " (L98), where "they" refers to a focal male, are misleading. "They" should refer to gene copies.

About relatedness coefficients: L83 informs us that they look at the degree of relatedness between two individuals sampled at random with replacement (i.e. including the focal). Then it is not clear to us how the authors arrive to the following interpretation for the first term of eq. (1) on L89-91: "the direct benefit enjoyed by the focal male from increased mating success $\frac{c}{2}$, [...] minus the mating success lost by his mate competitors $-\frac{c}{2}$, weighted by his relatedness to them r_{mm} " (where according to L83 r_{mm} is calculated with replacement). But shouldn't the relatedness between a focal male and its competitors (i.e. the other males in the patch) be calculated "without replacement"? Connected to this, it would be helpful if the authors were clearer about when (and how) the competition among males for siring, and male harm to females occur.

It is probably clear by now that at least to us, the connection between eq. (1) and the traits, their genetic basis, the life cycle and their fitness effects is difficult to grasp, even from reading the appendix. Given its current ambiguity and the seemingly false sense of generality that it gives, we would advise to drop the open model as such. The decomposition of the selection gradient according to the kin selection approach is of course extremely useful to interpret results. In our opinion, it would therefore be more powerful and easier to follow if the selection gradient was presented for a particular "closed model", which can be easily tied to clear biological processes, but in such a way that it is decomposed according to eq. (1) with all the relevant effects of interactions and competition among relatives to explain selection and models to come.

The closed models also suffer from a certain lack of clarity, in particular the soft selection cases where 0

Furthermore, because the two soft selection models are hard to understand, it's difficult to see how they're different from each other and therefore what the reason behind modelling both cases is. Similarly, no explanation is given as to why there are differences in harm potential between the two cases when s is intermediate as mentioned on L192-3. Additionally, the description of results given on L193-7 is also confusing. Firstly, it is stated that soft selection leads to reduction in harm potential on L193-4 but figures 2c-d appear to show the opposite pattern. Secondly, on L194-5 the authors say that the invariance of harm potential with respect to dispersal is

maintained under RDR but not DRD, but it is not clear where we are supposed to see this result as figures 2c-d and 3c-d both show output for a fixed level of dispersal.

Overall, much of the lack of clarity in this paper owes to the Appendix being frustratingly cursory, making rederivation and interpretation of the results near impossible. Many important equations are presented with minimal or no explanatory text (e.g. S9-12) and so are difficult to follow intuitively. To allow readers to follow the arguments behind the model, it is essential to provide more text that explain derivations, especially of the fitness equations.

Another issue for the paper's reproducibility is the weakness of the links between the Appendix and main text. For example the majority of the paper's results are presented numerically as figures, yet there is no explanation in either the main text figure legends or the Appendix as to how the figures are generated (e.g. which equations are used and what they are solved for in order to arrive at each of the curves). Each section of main text results should reference specific locations in the appendix where results are derived, and vice-versa, e.g. presumably the results in the section "Population viscosity drives intragenomic conflict between nuclear and cytoplasmic genes" are generated by analysing the closed model appendix equations and setting s and s to 0 and $dm = df$, but this needs to be clearly stated somewhere. In other words, we need to be able to recreate the paper's figures through a few guided substitutions, which is not the case currently.

From a more empirical perspective, an important question is what are the units of "harm potential" (H) in the context of the amount of harm that is apportioned to females? For example, what are the consequences for female fecundity of a 1 unit increase in H? To help, a more specific explanation of H after eq. 2 is needed - it appears that H gives the number of offspring of a random female in its patch that a male will sacrifice through harm in order to increase his own direct reproduction by one offspring, is this accurate? The ambiguity over how H is scaled also makes it difficult for the reader to interpret the effect sizes shown in the figures. In fact, a number of the reported effect sizes appear relatively small upon inspection. Specifically, the magnitude of the differences between nuclear genes in response to demography appears to be quite small in Figures 1b and especially 3a-b (which are scaled between 0.65 and 0.85) when plotted as $H/(1+H)$. This is not consistent with the relevant sections of the main text results. In addition, the potential for effect sizes makes it difficult to interpret the notion of hypo- and hyper- harming presented in the discussion and fig S1. To what degree would H increase/decrease in these cases and quantitatively what would the effects be for female fitness?

The issue over ambiguous effect sizes also draws into question the tone of the paper. Currently, the manuscript's abstract, discussion (and title!) rely on the demonstration that demography modulates conflict across genomic regions. However such a conclusion is not especially straightforward if many of the effect sizes are in fact small. In order to avoid overstating, we therefore suggest some changes to discussion and abstract are warranted, or clarification of the results to more convincingly show significant differences among genomic regions due to sex-specific demography.

In a similar vein, it is also clear from the analyses that selection on male harm is invariant across different nuclear regions for a number of common demographic cases (i.e. sex-congruent dispersal, reproductive skew, degree of soft selection). The consistency of this result across many genomic regions is not uninteresting and there should be a paragraph discussing the causes/implications of this in the discussion.

Minor Points

Main text:

L84 - should it be "gene flow"? And "i" and "j" be italicised here?

L84 - define the scale of competition.

L109 - unclear where "increasing the scale" means; as increases, more individuals compete locally, is this really an increase in scale?

L125 - state this is sex-independent dispersal.

L146 - reference a figure, in general the results section would benefit from more frequent and specific references to figures (e.g. which colour curves the reader should be looking at)

L166-7 - cite Taylor & Irwin, 2000 here.

L192 - word missing.

L207 - Could you please explain your result that “Typically, with male-biased dispersal, the potential for male harm is greatest for X chromosomes, and lowest for Y chromosomes ($\phi > \psi > \chi > \omega$). Conversely, when there is female-biased dispersal this ranking is usually reversed ($\psi < \phi < \chi < \omega$).” at greater length? Since females never carry a Y, then males never harm genetic relatives at the harming locus. However, they compete with genetic relatives at the harming locus when dispersal is limited, especially in males. So one could expect that the potential for harm is greater when dispersal is female biased.

L238 - “there” typo.

L274 - Could you be more specific about how you would test for the effects of non uniparental inheritance on male harm in these species? For instance what type of male harm has been observed in those species listed?

L366 - This “framework will guide” → This “model may help” perhaps?

Figure 3 Legend - What is dm?

Appendix:

L12 - Could you define absolute and relative fitness precisely? E.g. is $W_f \rightarrow f$ the expected total number of female offspring produced by a female offspring (including in philo- and parapatry)?

L21 - Could you give a few examples of how the breeding value would change according to looking at different genomic locations?

L33A - It’s confusing to use here if it also refers to reproductive skew later on.

L51A - space missing.

L48-57 - Make it clearer here what “survival” means (i.e. between-generations), as previously in the manuscript survival is also used in reference to reaching the adult mating pool as described in L3-4.

L59-60 - space missing (after DRD and RDR).

Eq. S2 - Please define W clearly.

Eq. S9 and S11: These need to be clearly explained.

Eq. S10 and S12: How are these equations connected to eq.S1? The open model was phrased as if it could be applied to any model but it seems difficult to express Eqs. S10 and S12 as eqs.S1.

L93 - Perhaps explain why alpha and beta are useful and how they allow you to consider different genetic loci.

Eq. S24 - Why are these probabilities useful ? i.e. when do you use them ?

L.147 - This section is much too terse. We need to know more clearly how we can reproduce your results.

Refs

Rousset, F. and Ronce, O., 2004. Inclusive fitness for traits affecting metapopulation demography. *Theoretical population biology*, 65(2), pp.127-141.

Taylor, P. D. 1992a. Altruism in viscous populations: an inclusive fitness model. *Evol. Ecol.* 6:352–356.

Taylor, P. D. 1992b. Inclusive fitness in a homogeneous environment. *Proc. R. Soc. Lond. B Biol. Sci.* 249:299–302.

Taylor, P. D., & Irwin, A. J. 2000. Overlapping generations can promote altruistic behavior. *Evolution*, 54(4), 1135-1141.

Wilson, D. S., G. B. Pollock, and L. A. Dugatkin. 1992. Can altruism evolve in a purely viscous population? *Evol. Ecol.* 6:331–341.

Author's Response to Decision Letter for (RSPB-2021-1461.R0)

See Appendix A.

RSPB-2021-2237.R0

Review form: Reviewer 2

Recommendation

Accept with minor revision (please list in comments)

Scientific importance: Is the manuscript an original and important contribution to its field?

Good

General interest: Is the paper of sufficient general interest?

Good

Quality of the paper: Is the overall quality of the paper suitable?

Good

Is the length of the paper justified?

Yes

Should the paper be seen by a specialist statistical reviewer?

No

Do you have any concerns about statistical analyses in this paper? If so, please specify them explicitly in your report.

No

It is a condition of publication that authors make their supporting data, code and materials available - either as supplementary material or hosted in an external repository. Please rate, if applicable, the supporting data on the following criteria.

Is it accessible?

N/A

Is it clear?

N/A

Is it adequate?

N/A

Do you have any ethical concerns with this paper?

No

Comments to the Author

We find the clarity of the revised manuscript, in particular with regard to the closed models, to be much improved. Enclosed are our remaining comments.

General comments

1) We regret that the authors are continuing with the open model, as this generates ambiguities regarding the life cycle which reduces transparency in the results. Furthermore, referring to the open models as more “general” is misleading because only a limited number of life cycles or life histories can in fact be represented this way (ie as in Frank, 1998, chapter 7). Ultimately, however, we leave the decision over to include the open model to the editor and the authors.

2) If included, there remains a lack of clarity over relationship between d and a is in the main text. Why can they be treated independently from one another, and which aspects of social evolution do they capture? Presumably while a captures kin competition, d affects which social partners feel the consequences of expressed harm?

3) With regards to the effect sizes, we appreciate the inclusion of a worked through example to demonstrate the magnitude of male trait size variation owing to H . However, the main text currently does not touch upon these results, and the relevant figures (S3-5) are not referenced at all (S5 is referenced in the discussion but we think this a mistake and the authors meant S6). We therefore recommend some sentences in the discussion or results that describes how variation in H relate to variation in trait.

4) Moreover, section SM§3 is also somewhat opaque. Which equations in Faria et al 2020 are being used here? What are the biological implications of assuming the functions given in S37-8 and what is the interpretation of κ here, as it appears to correspond roughly to the strength of mating competition? Furthermore, it is confusing to use κ here when it's also used under eq S9 to mean something around appendix L70.

5) There is still a lack of integration between the main text and supplement. For instance, the methods behind Figure 1a are still not referred to in the appendix. There are still only few references to appendix equations in the main text which makes reading difficult, at least for the more theory orientated reader.

Main text

L73-4 Are the references to life-cycle stages correct?

L80 What is class?

L114-121 Link better with existing studies, presumably these results are already known (e.g. Farria et al 2020), are the c terms the novel aspect added by this study?

L128 Unclear what this means

L139-143 Again as this is known, a citation is probably warranted.

L354 Presumably this should be S6?

Figures

How is Fig 1a generated? The details in SM2 only pertain to Fig 1b.

Appendix

L72-8 More detail is needed here. What does κ here represent biologically? What does it mean for gene effects to add or average across ploidy levels. Also, κ then appears to become a “ k ” on L77?

L249 Should this be main text figures 1a-b not 1-3?

References

Frank SA. 1998 Foundations of Social Evolution. Princeton University Press.

Decision letter (RSPB-2021-2237.R0)

24-Nov-2021

Dear Mr Hitchcock

I am pleased to inform you that your manuscript RSPB-2021-2237 entitled "Sex biased demography modulates male harm across the genome" has been accepted for publication in Proceedings B.

The referee and Associate Editor have recommended publication, but the referee has also suggested some minor revisions to your manuscript. Therefore, I invite you to respond to the referee's comments and revise your manuscript (I leave to your discretion the first comment with regard to the inclusion of the open model, but please address the referee's concerns as to how it is referred to). Because the schedule for publication is very tight, it is a condition of publication that you submit the revised version of your manuscript within 7 days. If you do not think you will be able to meet this date please let us know.

Yours sincerely,
Professor Loeske Kruuk
mailto:proceedingsb@royalsociety.org

Associate Editor
Board Member
Comments to Author:

The revised manuscript has dealt with the concerns raised previously by the reviewers and is much improved. As such, the reviewer has only some final suggests that should be implemented before the paper can be accepted. I appreciate that it may not be possible to address all of these due to space constraints. However, can the authors make sure that the appendix equations are appropriated referenced in the main text as well as the methods for Figure 1a and the supplementary analysis exploring the magnitude of male trait size variation owing to H.

Reviewer(s)¹ Comments to Author:

Referee: 2

Comments to the Author(s).

We find the clarity of the revised manuscript, in particular with regard to the closed models, to be much improved. Enclosed are our remaining comments.

General comments

1) We regret that the authors are continuing with the open model, as this generates ambiguities regarding the life cycle which reduces transparency in the results. Furthermore, referring to the open models as more “general” is misleading because only a limited number of life cycles or life histories can in fact be represented this way (ie as in Frank, 1998, chapter 7). Ultimately, however, we leave the decision over to include the open model to the editor and the authors.

2) If included, there remains a lack of clarity over relationship between d and a is in the main text. Why can they be treated independently from one another, and which aspects of social evolution do they capture? Presumably while a captures kin competition, d affects which social partners feel the consequences of expressed harm?

3) With regards to the effect sizes, we appreciate the inclusion of a worked through example to demonstrate the magnitude of male trait size variation owing to H . However, the main text currently does not touch upon these results, and the relevant figures (S3-5) are not referenced at all (S5 is referenced in the discussion but we think this a mistake and the authors meant S6). We therefore recommend some sentences in the discussion or results that describes how variation in H relate to variation in trait.

4) Moreover, section SM§3 is also somewhat opaque. Which equations in Faria et al 2020 are being used here? What are the biological implications of assuming the functions given in S37-8 and what is the interpretation of κ here, as it appears to correspond roughly to the strength of mating competition? Furthermore, it is confusing to use κ here when it's also used under eq S9 to mean something around appendix L70.

5) There is still a lack of integration between the main text and supplement. For instance, the methods behind Figure 1a are still not referred to in the appendix. There are still only few references to appendix equations in the main text which makes reading difficult, at least for the more theory orientated reader.

Main text

L73-4 Are the references to life-cycle stages correct?

L80 What is class?

L114-121 Link better with existing studies, presumably these results are already known (e.g. Farria et al 2020), are the c terms the novel aspect added by this study?

L128 Unclear what this means

L139-143 Again as this is known, a citation is probably warranted.

L354 Presumably this should be S6?

Figures

How is Fig 1a generated? The details in SM2 only pertain to Fig 1b.

Appendix

L72-8 More detail is needed here. What does kappa here represent biologically? What does it mean for gene effects to add or average across ploidy levels. Also, kappa then appears to become a "k" on L77?

L249 Should this be main text figures 1a-b not 1-3?

References

Frank SA. 1998 Foundations of Social Evolution. Princeton University Press.

Author's Response to Decision Letter for (RSPB-2021-2237.R0)

See Appendix B.

Decision letter (RSPB-2021-2237.R1)

26-Nov-2021

Dear Mr Hitchcock

I am pleased to inform you that your manuscript entitled "Sex-biased demography modulates male harm across the genome" has been accepted for publication in Proceedings B.

Data Accessibility section

Open Access

Paper charges

Sincerely,
Editor, Proceedings B
mailto: proceedingsb@royalsociety.org

Appendix A

Reply to Associate Editor

The manuscript was reviewed by two experts and myself. We all agreed that this is a high-quality paper that addresses an important gap in the literature. Here, the authors investigate how demography shapes the evolution of male harm traits, with a particular focus on how mode of inheritance influences the outcome. The topic is of broad interest to a range of topics, including evolution, sexual selection, kin selection, demography and genome evolution.

1. We thank the editor for their kind words, and we are really pleased that they feel our paper will be of interest to a wide range of researchers.

We have now made substantial changes to both the manuscript and the supplementary material in response to the reviewers' comments. However, we have been constrained in many places by the page limit, which we are aware is a hard one. In order to accommodate the recommended additions, we have now: (i) reduced our focus in the main text to only one soft selection scenario (RDR), discussing the other soft selection scenario in the Supplementary Material only; (ii) reduced the number of figure panels from 10 to 8, and combined these into a single figure rather than three separate ones; and (iii) removed a paragraph of the Discussion from our original submission (lines 347-359 of the original submission).

Additionally, we had intended to include a new table of results in the main text -- see point (4), below -- but owing to length constraints we have had to place this in the supplementary material instead (Table S2).

However, both reviewers raised a number of questions that need to be addressed.

Most importantly:

1) Both reviewers commented on the lack of clarity regarding the mathematics and assumptions of the model. It is clear that additional explanation and justification is required.

2. We are very grateful for the two reviewers for having spelled out their concerns so carefully. We have fully addressed their points – as detailed below – by expanding upon our account of the model and the methodology in the main text (lines 69-74, 81-88) as far as page limits have allowed us, and by greatly expanding our explanation and justification of the core assumptions of the model and the methodology in the supplementary material, particularly with regards to clarifying the life-cycle and how the fitness functions emerge from this (SM§1.1-1.2, Figure S1).

2) Both reviewers also commented on the disjunction between the supplementary material (where the mathematics of the model is outlined) and the main body of the manuscript. The paper would benefit from clearer signposting and a better explanation of the conditions of the model. Importantly, this should be understandable to non-theoreticians.

3. We have now added further detail in the main text on model assumptions that were previously discussed only in the supplementary material (lines 81-

88), and have now also introduced more thorough and detailed cross-referencing between main text and supplementary material to help readers navigate between the two (lines 75, 85, 163, 185, 202, 217, 364 & 367, Figure 1).

3) Related to the above points, the paper would benefit from a clearer explanation of the main predictions and results of the model. The first reviewer suggests including a table to summarise the main results. This may also help to deal with the second reviewer's concerns about effect sizes. This is an important point that should be addressed fully.

4. We agree that tabulating the key results is an excellent way of presenting this information, and we now include a new table (Table S2) for this purpose. In order to address Reviewer 2's concern about effect sizes we have (i) replotted Figures 1-3 (now combined into Figure 1) to make clear that the differences in the potential for harm are not necessarily small, (ii) developed a concrete illustrative example in the supplementary material to demonstrate how even small differences in the potential for harm may manifest as large differences in trait optima (SM§3), and (iii) added additional discussion on how even small differences in trait optima can drive intense coevolutionary dynamics (lines 358-361) – see point (16), below, for more details.

4) Finally, the second reviewer raises a number of thoughtful questions about specific assumptions and parameters of the model that should be addressed.

5. We have fully addressed Reviewer 2's concerns in the revised paper, as spelled out below. In particular, we have added substantially more detail and clarification to the supplementary material (in particular SM§1.1-1.2 & 2.1-2.4) regarding the assumptions and parameters of the model, which makes our results easier to interpret and replicate.

Reply to Reviewer 1

This paper presents an interesting gap in the literature and addressed it in a theoretical manor which seemed very suitable. They investigate how different modes of inheritance (via autosomes, sex chromosomes, or cytoplasmic elements) have potentially different evolutionary outcomes due to differences in demography. They approach the problem first with a non-explicit mathematical model and then expand to an explicit version of the model, allowing them to investigate more questions. I thought the paper was written well and I appreciated the thorough approach to the relevant literature – well done!

6. We are grateful for these kind words and are really pleased that the reviewer enjoyed our paper.

Whilst I thought the paper was well written, I have some points that I think need to be addressed to help make the manuscript as transparent and clear as possible. These points are as follows –

- Virtually all the mathematics was stored in the supplementary material. I do agree that it makes sense to store a lot of it in there, as there are plenty of equations in there which may be daunting to a casual reader. However, I think might be helpful for the audience if you could add some discussion about how you are deriving these conditions for increasing male*

harm in a general sense. For example, describing the kin-selection approach of Taylor and Frank you mention on line 77. I think this might make the results clearer without making the reader have to go into the supplementary information to understand how the method works.

7. We have now added more detail on our methodological approach in the main text to help give the reader a better understanding of our analysis and thus the meaning of the invasion conditions that we have derived (lines 81-88). We have also more comprehensively cross-referenced to the supplementary material throughout the main text in order to help the reader navigate between the two (lines 75, 85, 163, 185, 202, 217, 364 & 367, Figure 1).

• As I have understood it, the authors have only considered male harm in the context of affecting the fecundity of the female he mates with. This seems to ignore the potential for a mortality effect on females due to male harm, which is likely. I think there should be some discussion of how such a mortality effect might affect their results even if the authors do not investigate this as thoroughly as the fecundity effect.

8. We have indeed focused on fecundity – rather than survival – effects, mostly for reasons of simplicity but also to facilitate comparison with previous analyses of sexual conflict (e.g. Rankin 2015, Faria et al. 2015, 2020). Whilst survival effects would be interesting and worthwhile to investigate, we feel that properly dealing with this addition is beyond the scope of the present paper. This is in part due the fact that specific results will likely differ depending on assumptions as to the timing of mortality during the life-cycle (e.g. pre- versus post-fecundity), and how mortality effects alter the number of available breeding spots (e.g. soft versus hard selection). Accordingly, a properly robust account of mortality effects would require the development of a whole suite of new models, and there is simply not space to do this justice within the page limits for this paper. Nonetheless, the models we have analysed do already capture some survival effects, as reductions in female reproductive success during the reproductive portion of the life-cycle could be interpreted as owing to premature deaths during this interval. We have now added some discussion on this possible interpretation (lines 380-385).

• It was not completely clear to me what the authors meant by population regulation. For example, is that density-dependent mortality, a generational mortality, or some other mechanism. I was unsure exactly what the authors meant and because of that its harder to understand the significance of the differences between the DRD and RDR selection systems. It would be helpful if the authors could clarify exactly what they mean and the subsequent significance for the different selection regimes.

9. We agree that the explanation given in the previous version was not sufficiently clear and we have remedied this in the new version of the paper. What we mean by population regulation is the mechanism by which population size is held constant across generations, and in our case ensures that there are always n_m adult males and n_f adult females in every patch at the start of each iteration of the life cycle. We have now added some extra discussion to clarify this point (lines 69-74 & 203-204). The timing of population regulation within the life-cycle is important as it modulates the intensity of kin competition, and we have provided more discussion in the supplementary

material to clarify the distinction between the two life-cycles that we consider in our investigation of the role of soft selection (SM§2.1, Figure S1).

• The authors present a lot of results which is great, but it does make finding the key results and other results one may be interested in harder to identify. I think it would be helpful if the authors could add something like a table to summarise all the results.

10. We agree that a table is provides an excellent means of summarising the key results and implemented this suggestion in the form of Table S2. We feel that this makes the results more accessible to the reader and facilitates comparison between them.

• I personally think the mathematical approach used is suitable and makes sense, but my viewpoint is that of a mathematician. In the interest of a more general reader, I think it could be helpful to briefly mention some of the strengths and weaknesses of the approach used.

11. We have now added additional details concerning the assumptions behind the Taylor & Frank methodology (specifically: weak selection, additivity, and vanishingly rare genetic variation) and have added caveats about the scenarios in which our results are likely to be less informative (lines 85-88).

I have a few more comments about the supplementary material that I think are just as important to address, since that is where the technical detail is presented. In general, the points can be summarised as wanting more justification and description of the function choice, but I will outline a couple of specific cases below to try and make my point clear.

12. We have now expanded the description of the life cycle in the supplementary material (SM§1.1), including an additional supplementary figure (Figure S1). We have also more explicitly divided up the fitness functions to make clearer the distinction between an adult's fecundity versus the probability of an offspring attaining a breeding spot (SM§1.2, SM§2.1), which makes the structure of the fitness functions and their derivation easier to understand.

• In the description of Table S1, $a_{\{f\}}$ and $a_{\{m\}}$ are defined for both the DRD and RDR selection models. I would like to see some explanations as to why they take these forms and why these forms are biologically suitable definitions.

13. We have now expanded our discussion on the DRD and RDR mechanisms of population regulation (SM§2.1), and more clearly outlined how the fitness functions emerge from the life-cycle (Figure S1b).

• Likewise, equations (S9a-f), (S11a-f) give definitions for the relative fitness, however it does not seem clear why they are of this form. Have you derived these from first principles, chosen them arbitrarily, or followed others work? Whichever it is, it would be nice to highlight where these equations have come from, and why they are suitable functional forms.

14. The relative-fitness expressions are based on those of Faria et al (2020). However, whereas Faria et al were able to average across males and females to give a single intensity-of-local-competition parameter a , on account of their

focus on sexually-symmetrical autosomal inheritance, this is not possible under the wider set of inheritance systems we consider in this paper, and hence we are required to describe separate, sex-specific parameters. We now make this clearer, along with spelling out how relative fitness is defined as the ratio of absolute fitness (attained via a particular route in the life-cycle, i.e., survival versus reproduction, and sons versus daughters) and the population-average of absolute fitness (attained via this same class transition; SM§1.2).

Reply to Reviewer 2

Summary

In this paper, the authors investigate the evolution of male harm (a form of interlocus sexual conflict where males increase their siring success at the expense of females' fecundity) across different genomic locations in group structured populations such that individuals interact and compete among relatives. They do so by analysing the conditions that favour the invasion of an allele at a locus coding for male harm when this locus can sit on an autosome, a X-, Z-, or Y-chromosome, or a cytoplasmic genomic element. A number of demographic parameters that are relevant to group-structured populations are also allowed to vary (including in a sex-specific manner), such as dispersal, adult survival, degree of reproductive skew, and whether regulation occurs before or after dispersal. They find that the different nuclear genomic locations have exactly the same propensity to favour male harm when demographic parameters are equal across males and females (e.g., male and female juveniles disperse at the same rate). By contrast, the potential for cytoplasmic genomic elements to harbour male harm alleles can be sensitive to demography (provided inheritance is not strictly uniparental). Where nuclear genomic locations do differ in their potential for male harm is where demographic parameters differ among the sexes, such as dispersal rate or timing. They use their result to discuss the genomic distribution of male harm as well as potential intra-genomic conflict for male harm.

Overall, we have a positive impression of this manuscript. The introduction and discussion are well-written and should eventually be useful for future discussions on the genomic location of alleles influencing male harm. The breadth of scenarios investigated, in particular, is commendable.

15. We are grateful for these kind words and are pleased that the reviewer enjoyed our paper.

There are however also a number of major issues with the current version of this manuscript. Two seem particularly important. First, there is a significant lack of clarity about the assumptions underlying some of the models. This makes following the authors' arguments challenging at times, even for theoretical biologists, and ultimately clouds the biological interpretation of the results. Second, and probably more troublesome from an empirical perspective, is that many of the reported effect sizes seem small. This in itself is not a problem (at least to us) but it sits at odds with the language employed in the main text, with as a consequence the main claims of the paper looking a bit overplayed. We expand on these points below.

16. We have now added a great deal more detail into the supplementary material on the assumptions of the model, the basis for the fitness functions

and the methodological approach that we have employed (SM§1.1-3, SM§2.1, SM§2.4, Figure S1). We feel these additions will make it easier for the reader to interpret the invasion conditions and replicate the results we have derived.

We also appreciate that some of the effect sizes appeared to be quite small in the figures given in the original version of the paper, and thus differences amongst nuclear genes may have seemed as though they would be difficult to detect empirically. However, we feel that these differences are empirically relevant, for the following reasons:

- **Firstly, the differences in potential for harm may have appeared smaller than they actually were, owing to the way in which they were scaled such that all values from zero to infinity would fit along the vertical axis. We have resolved this issue by adopting an alternative scaling in Figure 1 in the revised version of the paper.**
- **Secondly, even seemingly small differences in potential for harm can translate into large differences in trait optima, depending upon the functions chosen to represent the marginal benefits and costs. To illustrate this point, in the revised paper we have now added a concrete example involving the particular male and female fitness effects assumed by Faria et al. (2020). Whilst the magnitude of these differences in trait optima varies (and depend on the extent of sex bias), under reasonable assumptions it can be substantial (~10% difference between X versus Y chromosomes) and hence empirically detectable at least in some natural and/or experimental systems (see SM§3).**
- **Thirdly, even small differences in trait optima may drive escalating intragenomic conflicts with empirically detectable consequences. For example, biased trait control emerging as a consequence of intragenomic conflict could be detectable via crosses that disturb the delicate balance of control, resulting in strikingly maladapted phenotypes. We now provide explicit discussion of such possibilities in the revised paper (lines 358-361, SM§4).**

Major Points

The assumptions behind the “open model” are too ambiguous. The main text suggests that eq. (1) applies to a broad set of models whose assumptions are those mentioned on L68-71. First it is not clear to us what is meant by “with a proportion of the resulting competition occurring locally, in a sex-specific manner”. What is a proportion of competition exactly? It is the ratio of which quantities? In particular, how is it connected to individual processes of competition for resources, competition for breeding spots and dispersal?

17. We have now added further detail in the main text to clarify the particular life-cycle structure encapsulated by the open model (lines 69-74) and have expanded upon this point in the supplementary material (SM§1.1, Figure S1). We feel that these clarifications make the model more straightforward to understand and the analysis easier to replicate.

With regards to the statement “with a proportion of the resulting competition occurring locally, in a sex-specific manner”, the idea here is that fitness is always relative, and hence expressed relative to some set of competitors, with some proportion of the competition occurring locally (within the patch) and the remainder occurring globally (outwith the patch), and that these proportions can differ between the sexes – for example, males might compete only with local males, and females might compete more globally. We have clarified this conceptualisation of the degree of local competition – making more explicit its origins in Frank’s (1998) *Foundations of Social Evolution* – in the revised paper (lines 97-99).

We do feel that the open model’s utility is precisely that it avoids specifying the exact mechanism by which kin competition arises, because therein lies its generality and hence its ability to connect and synthesize across a vast set of different model scenarios. Using the “scale of competition” parameter a (Frank 1998) allows to us capture the intensity of kin competition without restricting ourselves to a particular demographic scenario. Thus, the parameter a applies to a diverse set of demographics, including not only the demographically explicit (closed) models that we have analysed, but also demographics outwith our analysis, such as budding dispersal (Gardner & West 2006, Faria et al. 2020). This is similar to the utility of leaving the ecological and demographic parameters underpinning relatedness r unspecified, which allows us to understand the influence of relatedness *per se* on the evolution of harming traits. We have added some words on the utility of open models in the revised paper (lines 129-131).

Second, are L68-71 really the only assumptions required for eq. (1)? Shouldn’t there at least also be random mating (and interactions among the sexes) within patches? This should be mentioned given that the paper is about the evolution of such interactions. Shouldn’t dispersal be uniform and random among patches? Shouldn’t also the number of adult males and females be fixed and constant in all patches (otherwise the selection gradient i.e. eq. (1), should incorporate the effects of traits on local demography, see Rousset & Ronce, 2004 for inclusive fitness for traits affecting metapopulation demography). Actually, don’t equations S1 in the appendix suggest that the sex ratio among adults is 1:1 ($n_m = n_f$) as equations S1 tell us that in the absence of genetic variation (when $x=y=z$) a male produces on average one female and one female produces one male? This is a non-exhaustive list. As is expected from a theory paper, the authors should be clear about what types of traits and populations can be understood with eq. (1).

18. We agree that in the previous version of the paper, some of these details were unclear, and certain assumptions remained implicit. We have now made these assumptions more explicit, both in the supplementary material (e.g. that dispersal is uniform and random among patches; SM§2.1), and also in the main text (e.g. the number of adult females and males in each patch is fixed; lines 69-70). More generally, we have now added more detail about the assumptions underlying the open and closed models (lines 69-74).

There is no requirement that the sex ratio of adult males and females be 1:1. Whilst this simplifying assumption is made in our plots and in some of our equations for the purpose of illustration, more generally our analysis does

allow for arbitrary sex ratio bias. We think that perhaps our expressions for relative fitness (which is, by definition, equal to 1 on average) were being misinterpreted as expressions for absolute fitness (for which the average value may be different from 1). We have now included some additional discussion of the effect of patch size in the main text (lines 201-202, 255-259) and a new figure to demonstrate how the number of males (n_m) and females (n_f) alter the potential for harm in the supplementary material (Figure S2, Table S2), to provide more emphasis of the model's scope for biased sex ratios.

Another confusing aspect of eq. (1) is how quantities are verbally defined. From the introduction and the rest of the manuscript, we expect all quantities to be defined at the genetic level in eq. (1) as eq. (1) is supposed to be able to consider alleles that can sit on different chromosomes with different patterns of segregation and sex-specificities. In other words, we expect the fitness effects (B and C), the relatedness coefficients (r), reproductive values (c) and competition effects to be defined in terms of a focal gene rather than a focal individual. How do you navigate between the different genetic models from eq. (1) e.g. how do you go from an autosome to a Y-chromosome? Is that all contained in the class reproductive values? If so, statements like “the reproductive value they would have accrued through their daughters $\phi_{m \rightarrow f}$ and sons $\phi_{m \rightarrow m}$ ” (L98), where “they” refers to a focal male, are misleading. “They” should refer to gene copies.

19. Our analysis is framed in terms of the marginal costs and benefits experienced by a focal individual, rather than those experienced by a focal gene. We analyse how a change in the individual's behaviour may be favoured or disfavoured depending on the inheritance system of a controlling gene. The differences between inheritance systems are indeed captured by the class reproductive values – and also by the relatedness coefficients – that weight the marginal costs and benefits in the calculation of the overall action of natural selection. We have added some additional discussion to make these points clearer (lines 91-93 & 133-134).

About relatedness coefficients: L83 informs us that they look at the degree of relatedness between two individuals sampled at random with replacement (i.e. including the focal). Then it is not clear to us how the authors arrive to the following interpretation for the first term of eq. (1) on L89-91: “the direct benefit enjoyed by the focal male from increased mating success $\phi_{m \rightarrow f}$ [...] minus the mating success lost by his mate competitors — $\phi_{m \rightarrow m}$ weighted by his relatedness to them r_{mm} ” (where according to L83 r_{mm} is calculated with replacement). But shouldn't the relatedness between a focal male and its competitors (i.e. the other males in the patch) be calculated “without replacement”? Connected to this, it would be helpful if the authors were clearer about when (and how) the competition among males for siring, and male harm to females occur.

20. We agree that this phrasing was unclear and we have now changed the wording to avoid ambiguity (lines 104-105). Instead of referring to the male's “mate competitors” we now refer to “the average male on his patch”, and we make clear that it is the “whole-group” sense of relatedness (*sensu* Pepper 2000) that is being employed here (lines 95). We have also clarified the assumptions concerning the life-cycle (lines 69-74, SM§1.1, Figure S1), and thus the timing of competition amongst males and harming of females (lines 77-78).

It is probably clear by now that at least to us, the connection between eq. (1) and the traits, their genetic basis, the life cycle and their fitness effects is difficult to grasp, even from reading the appendix. Given its current ambiguity and the seemingly false sense of generality that it gives, we would advise to drop the open model as such. The decomposition of the selection gradient according to the kin selection approach is of course extremely useful to interpret results. In our opinion, it would therefore be more powerful and easier to follow if the selection gradient was presented for a particular “closed model”, which can be easily tied to clear biological processes, but in such a way that it is decomposed according to eq. (1) with all the relevant effects of interactions and competition among relatives to explain selection and models to come.

21. We have now clarified our description of the open model, both in terms of the underlying life-cycle (lines 69-74, SM§1.1, Figure S1), and also how the fitness functions emerge from the life-cycle (SM§1.2), to remove ambiguity.

We feel it is important to retain the open model for two reasons. Firstly, it captures demographic structures outwith the ones we have analysed explicitly, for example budding dispersal (Gardner & West 2006, Faria et al. 2020), and therefore achieves a level of generality that would not be possible with a closed-model approach. Secondly, as the reviewer points out, it provides a useful way to conceptualise the core forces acting on the evolution of harm – i.e. in relation to reproductive value, relatedness, and kin competition – and thus better understand what is going on in closed models wherein closed expressions can often be difficult to interpret.

Put another way, whereas closed models give us the *what*, open models give us the *why*. We understand that open models are not to everyone’s tastes, and for the benefit of such readers we have undertaken the closed modelling. But, equally, other readers do prefer open models, because they provide more general insights, and for the benefit of these readers we have undertaken the open modelling. Cooper et al (2018, “Modelling relatedness and demography in social evolution”, *Evol Lett*) have recently provided a full discussion of the benefits of combining both open and closed model approaches. We now give some further clarification of these points in the revised paper (lines 129-132).

The closed models also suffer from a certain lack of clarity, in particular the soft selection cases where 0

22. We think some text has been omitted from the reviewer’s comment here. Nevertheless, we get the gist of it. We have now only focused on one soft selection scenario in the main text (RDR), discussing the other in supplementary material, where we have clarified our description (Figure S1 & SM§2.1).

Furthermore, because the two soft selection models are hard to understand, it’s difficult to see how they’re different from each other and therefore what the reason behind modelling both cases is. Similarly, no explanation is given as to why there are differences in harm potential between the two cases when s is intermediate as mentioned on L192-3. Additionally, the description of results given on L193-7 is also confusing. Firstly, it is stated that soft

selection leads to reduction in harm potential on L193-4 but figures 2c-d appear to show the opposite pattern. Secondly, on L194-5 the authors say that the invariance of harm potential with respect to dispersal is maintained under RDR but not DRD, but it is not clear where we are supposed to see this result as figures 2c-d and 3c-d both show output for a fixed level of dispersal.

23. We agree that the distinction between the two soft selection models is subtle and was not explained as clearly as it could have been. To address this, we have now focused on only one soft selection scenario in the main text (RDR), leaving discussion of the DRD scenario to the supplementary material. There we provide further clarification of the distinction between the two (in particular, explaining that there is a greater degree of kin competition for a given value of s under RDR than under DRD, such that RDR is more favourable to harm).

The motivation for studying both of these cases was that, when considering how to implement soft selection in our model, we realised that previous models (e.g. Gardner & West 2006 and Debarre & Gandon 2011) had used different approaches to scale between the extremes of hard and soft selection. We found that whilst in one approach (RDR) the invariance with respect to dispersal holds, in the other (DRD) it does not, and thus the potential for harm differed between the two approaches to modelling soft selection. We felt this worth highlighting, and of potential interest not only to theoreticians but also to empiricists (particularly those undertaking experimental work) whose choices as to the implementation of soft selection may be affected by the structure of their life-cycles of interest.

Overall, much of the lack of clarity in this paper owes to the Appendix being frustratingly cursory, making rederivation and interpretation of the results near impossible. Many important equations are presented with minimal or no explanatory text (e.g. S9-12) and so are difficult to follow intuitively. To allow readers to follow the arguments behind the model, it is essential to provide more text that explain derivations, especially of the fitness equations.

24. We agree that the previous version was too terse. To do fix this, and improve the replicability of our analysis, we have now added further diagrams describing the various life-cycles considered (Figure S1), and we have added more explanation as to how the core fitness functions emerge from these life-cycle (SM§1.1-1.2), and how solutions for the closed models are generated (SM§2.4).

Another issue for the paper's reproducibility is the weakness of the links between the Appendix and main text. For example the majority of the paper's results are presented numerically as figures, yet there is no explanation in either the main text figure legends or the Appendix as to how the figures are generated (e.g. which equations are used and what they are solved for in order to arrive at each of the curves). Each section of main text results should reference specific locations in the appendix where results are derived, and vice-versa, e.g. presumably the results in the section "Population viscosity drives intragenomic conflict between nuclear and cytoplasmic genes" are generated by analysing the closed model appendix equations and setting s to 0 and $dm = df$, but this needs to be clearly stated

somewhere. In other words, we need to be able to recreate the paper's figures through a few guided substitutions, which is not the case currently.

25. We have now made the analysis of the closed models clearer in the supplementary material (SM§2.4), giving explicit equations for the scenarios with no sex bias for both nuclear and cytoplasmic genes, making clear how the equations for the sex-biased scenarios are generated (SM§2.4), and cross-referencing with this section in our figure (Figure 1).

From a more empirical perspective, an important question is what are the units of “harm potential” (H) in the context of the amount of harm that is apportioned to females? For example, what are the consequences for female fecundity of a 1 unit increase in H ? To help, a more specific explanation of H after eq. 2 is needed - it appears that H gives the number of offspring of a random female in its patch that a male will sacrifice through harm in order to increase his own direct reproduction by one offspring, is this accurate? The ambiguity over how H is scaled also makes it difficult for the reader to interpret the effect sizes shown in the figures. In fact, a number of the reported effect sizes appear relatively small upon inspection. Specifically, the magnitude of the differences between nuclear genes in response to demography appears to be quite small in Figures 1b and especially 3a-b (which are scaled between 0.65 and 0.85) when plotted as $H/(1+H)$. This is not consistent with the relevant sections of the main text results. In addition, the potential for effect sizes makes it difficult to interpret the notion of hypo- and hyper- harming presented in the discussion and fig S1. To what degree would H increase/decrease in these cases and quantitatively what would the effects be for female fitness?

26. We agree that the explanation of the potential for harm was quite terse. We have now emphasised that B and C are scaled marginal costs and benefits (lines 91-92), as thus they are both in units of h^{-1} , where h is the unit of male harm. Accordingly, H , as the threshold ratio of C/B , is a dimensionless quantity, which we have now stated in the main text (line 116). We have also added some further discussion of this in the supplementary material when we initially introduce the potential for harm (SM§1.3).

On the issue of apparently small effect sizes, see point (16), above.

There is no direct link between the magnitude of the differences in the potential for harm across different parts of the genome (beyond it being nonzero) and the extent of the hypo- and hyper-harming phenotypes considered in the *Discussion* and in Figure S1 (now Figure S6). These maladaptive – yet qualitatively predictable – phenotypes do not correspond to the optima for any of the genomic elements considered in our analysis, but rather represent a breakdown owing to hybridization (or other such events). Extreme phenotypes are expected here on account of the intragenomic conflict driving escalation of gene effects that would usually balance out to give a normal phenotype but which take the phenotype in drastic directions if there is any disruption to the balance of power (just as in a tug of war, wherein the players find themselves violently wrenched backwards, well beyond where they intended to be, if the rope suddenly snaps). It is challenging to make quantitative predictions as to how differences in the potential for harm would translate into the magnitude of the hypo- and hyper-harming phenotypes; this

would strongly depend upon further assumptions about the particularities of both the biology of harm and the genetics underpinning the trait. Nonetheless, the qualitative pattern we have outlined is robust, and the basic principle of uncovering signatures of intragenomic conflicts through crosses has been successfully applied to understand a range of empirical systems and intragenomic conflicts of interest. We have now added further discussion on this issue in SM§4.

The issue over ambiguous effect sizes also draws into question the tone of the paper. Currently, the manuscript's abstract, discussion (and title!) rely on the demonstration that demography modulates conflict across genomic regions. However such a conclusion is not especially straightforward if many of the effect sizes are in fact small. In order to avoid overstating, we therefore suggest some changes to discussion and abstract are warranted, or clarification of the results to more convincingly show significant differences among genomic regions due to sex-specific demography.

27. As detailed in point (16), above, we feel that the predictions are empirically testable through a range of different avenues, including in relation to the study of extreme phenotypes emerging as a consequence of crosses.

In a similar vein, it is also clear from the analyses that selection on male harm is invariant across different nuclear regions for a number of common demographic cases (i.e. sex-congruent dispersal, reproductive skew, degree of soft selection). The consistency of this result across many genomics regions is not uninteresting and there should be a paragraph discussing the causes/implications of this in the discussion.

28. We agree that this result is interesting. However, due to the page limits, we feel that a whole extra paragraph is unwarranted. We have instead added a few sentences explaining the context of these results a little more, and drawing links to analogous results obtained by other analyses (lines 320-327).

Minor Points

Main text:

L84 - should it be "gene flow"? And "i" and "j" be italicised here?

29. Fixed.

L84 - define the scale of competition.

30. We have now added an additional definition here (lines 97-99), and we have also more thoroughly expanded upon this in the supplementary material in relation to the description of the open model and corresponding fitness functions (SM§1.2).

L109 - unclear where "increasing the scale" means; as increases, more individuals compete locally, is this really an increase in scale?

31. The phrase "scale of competition" was introduced by Frank (1998) to describe the extent to which competition is occurring locally. We agree that it

is potentially confusing that increasing the scale of competition pertains to an increase in the extent of local, rather than global, competition. For this reason, we have now reworded these passages throughout the revised paper so as to instead refer to “the intensity of kin competition”, as this eliminates any ambiguity (lines 50, 60, 96, 124, 129, 135, 183, 196, 220 & 238).

L125 - state this is sex-independent dispersal.

32. Done (line 147).

L146 - reference a figure, in general the results section would benefit from more frequent and specific references to figures (e.g. which colour curves the reader should be looking at)

33. Done (lines 162,168).

L166-7 - cite Taylor & Irwin, 2000 here.

34. Done.

L192 - word missing.

35. Fixed.

L207 - Could you please explain your result that “Typically, with male-biased dispersal, the potential for male harm is greatest for X chromosomes, and lowest for Y chromosomes ($\rho_{XX} > \rho_{XY} > \rho_{YY}$) . Conversely, when there is female-biased dispersal this ranking is usually reversed ($\rho_{XX} < \rho_{XY} < \rho_{YY}$) .” at greater length? Since females never carry a Y, then males never harm genetic relatives at the harming locus. However, they compete with genetic relatives at the harming locus when dispersal is limited, especially in males. So one could expect that the potential for harm is greater when dispersal is female biased.

36. In fact, in this scenario, the potential for harm doesn't change at all for the Y chromosome. As the reviewer says, it is unaffected by the dispersal – or any other behaviour – of females. But, contrary to what the reviewer suggests, it is also unaffected by the dispersal of males. This is because, although reducing the rate of male dispersal does lead to an increase in relatedness, as described by the reviewer, it also leads to an increase in kin competition, and these two effects exactly cancel (à la Taylor 1992).

The potentials for harm for the other genomic elements are affected by dispersal. As male dispersal decreases (Figure 1b), then kin competition increases, and relatedness increases too. However, all of the kin competition increase is in relation to males, whereas the relatedness increase affects both matrilineal and patrilineal relatedness (although patrilineal relatedness more so). Thus, Taylor's cancellation does not obtain and the X chromosome – which is related more so matrilineally, and transmitted disproportionately through daughters – experiences a greater increase in relatedness relative to kin competition in comparison to the other genomic elements.

We have added some extra words to clarify this point in the revised paper (lines 237-241).

L238 - “there” typo.

37. Fixed.

L274 - Could you be more specific about how you would test for the effects of non uniparental inheritance on male harm in these species? For instance what type of male harm has been observed in those species listed?

38. In general, we may expect species with non-matrilineal inheritance of cytoplasmic genes to show higher harm than those species with strict matrilineal inheritance of cytoplasmic genes. However, this is only likely to be detectable in cases where such genes do have an influence over the harming phenotype. Whilst harm is known to occur in some of the mentioned species (e.g. persistent courtship of males in tsetse flies; Clutton-Brock & Langley 1997), it is not clear the role that the particular cytoplasmic genes play in these forms of harm. However, we have now included a further example of biparental inheritance in the sigma clade of rhabdoviruses, of which there is some evidence of their having a direct effect on male reproductive success, and thus may provide a potential model system for investigating the predictions we’ve outlined here (lines 303-312).

L366 - This “framework will guide” → This “model may help” perhaps?

39. Done.

Figure 3 Legend - What is d_m ?

40. We now stipulate that $d_m = 1/2$ for these plots.

Appendix:

L12 - Could you define absolute and relative fitness precisely? E.g. is $W_f \rightarrow f$ the expected total number of female offspring produced by a female offspring (including in philo- and parapatry)?

41. We have now made the definitions for absolute and relative fitness more precise. The absolute fitness refers to the absolute number of adults (of the appropriate sex) in the next iteration of the life cycle assigned parentage to our focal individual (either through survival or reproduction). The relative fitness is the absolute fitness, divided by the average absolute fitness of an individual in that same class. These definitions are now given more clearly in SM§1.2.

L21 - Could you give a few examples of how the breeding value would change according to looking at different genomic locations?

42. The breeding value of the individual at this locus G may change depending on assumptions about how the number of individual gene copies determine G ,

and thus how the breeding value may change depending on the ploidy of male, i.e., the distinction between “adding” versus “averaging” gene effects (Frank 2003, Gardner 2012). If gene effects are “adding” then the breeding value of diploid males at a particular locus may be expected to be twice that of haploid males at the same locus. In our model this ultimately does not impact the results, as we assume an exclusively male limited trait, such that this detail would simply scale the marginal fitness costs and benefits by a constant amount. We have now made this clearer by explicitly including the parameter κ_m in this section (SM§1.3).

L33A - It's confusing to use here if it also refers to reproductive skew later on.

43. We have now changed this to ζ .

L51A - space missing.

44. Fixed.

L48-57 - Make it clearer here what “survival” means (i.e. between-generations), as previously in the manuscript survival is also used in reference to reaching the adult mating pool as described in L3-4.

45. We have now made clearer – in both the main text (line 69-74, 187) and in the supplementary material (SM§1.1, Figure S1) – the distinction between “between generation mortality” of adults and “within generation mortality” of juveniles as a result of competition for breeding spots.

L59-60 - space missing (after DRD and RDR).

46. Fixed.

Eq. S2 - Please define W clearly.

47. We have now made clearer (in SM§1.2) the definitions of absolute (w) and relative (W) fitness, and have now given explicit expressions for absolute fitness in the open model section.

Eq. S9 and S11: These need to be clearly explained.

48. We have now made clearer in our description of the life-cycle where the fitness functions emerge from (Figure S1, SM§1.1), and have also split up the absolute fitness functions more clearly to delineate the change in absolute fecundity versus the change in the probability of obtaining a breeding spot (SM§1.2).

Eq. S10 and S12: How are these equations connected to eq.S1? The open model was phrased as if it could be applied to any model but it seems difficult to express Eqs. S10 and S12 as eqs.S1.

49. Equations S10 (now S14) and S12 (now S17) are not directly derived from the equations in the open model (now S4). Instead the expressions for the marginal fitness effects can be recovered from those generated from equations S4 by setting a_f and a_m to particular values. This illustrates how the open model can capture a wide variety of other demographic settings within the kin competition parameters. We now make this clearer in Table S1 and SM§2.1.

L93 - Perhaps explain why alpha and beta are useful and how they allow you to consider different genetic loci.

50. The parameters α (the probability that a gene sampled in a female was inherited from a female) and β (the probability that a gene sampled in a male was inherited from a male), provide succinct ways for us to write out a single set of recursion equations, which can then recover the consanguinities of various inheritance systems. For example, autosomal ($\alpha = 1/2, \beta = 1/2$), X-chromosome ($\alpha = 1/2, \beta = 0$), Z chromosome ($\alpha = 0, \beta = 1/2$), cytoplasmic elements ($\alpha = 1-\lambda, \beta = \lambda$), and Y chromosomes ($\alpha = 0, \beta = 1$) can all be neatly recovered by making these substitutions. (Note that, in the case of the Y chromosome, this mathematically treats females as though they have a paternally inherited Y chromosome, albeit one that is neither transmitted nor expressed, and is thus equivalent to being absent altogether.) We have now expanded SM§2.2 to include this information.

Eq. S24 - Why are these probabilities useful ? i.e. when do you use them ?

51. The reproductive values provide the appropriate weights on the different marginal fitness effects, as these weights describe the relative fraction of the ancestry that is flowing through these routes in the life-cycle. We have now made clearer that these are the class reproductive values that are substituted into equation S10 to obtain the results for the closed models.

L.147 - This section is much too terse. We need to know more clearly how we can reproduce your results.

52. We have now expanded SM§2.4 to describe how the marginal fitness effects, consanguinities, and reproductive values calculated in SM§2.1-SM§2.3 combine with equation S10, to give the condition for increase in the closed models, and how this is then used to obtain the potential for harm. We have now also included the explicit equations for cytoplasmic elements here (equations S34-36). We feel this makes it easier for the reader to replicate our findings.

Refs

Rousset, F. and Ronce, O., 2004. Inclusive fitness for traits affecting metapopulation demography. Theoretical population biology, 65(2), pp.127-141.

Taylor, P. D. 1992a. Altruism in viscous populations: an inclusive fitness model. Evol. Ecol. 6:352-356.

Taylor, P. D. 1992b. Inclusive fitness in a homogeneous environment. Proc. R. Soc. Lond. B Biol. Sci. 249:299-302.

Taylor, P. D., & Irwin, A. J. 2000. Overlapping generations can promote altruistic behavior. Evolution, 54(4), 1135-1141.

Wilson, D. S., G. B. Pollock, and L. A. Dugatkin. 1992. Can altruism evolve in a purely viscous population? Evol. Ecol. 6:331-341.

Appendix B

Reply to Editor

I am pleased to inform you that your manuscript RSPB-2021-2237 entitled "Sex biased demography modulates male harm across the genome" has been accepted for publication in Proceedings B.

The referee and Associate Editor have recommended publication, but the referee has also suggested some minor revisions to your manuscript. Therefore, I invite you to respond to the referee's comments and revise your manuscript (I leave to your discretion the first comment with regard to the inclusion of the open model, but please address the referee's concerns as to how it is referred to). Because the schedule for publication is very tight, it is a condition of publication that you submit the revised version of your manuscript within 7 days. If you do not think you will be able to meet this date please let us know.

...

Yours sincerely,

*Professor Loeske Kruuk
mailto:proceedingsb@royalsociety.org*

1. We are delighted with the positive decision on our paper. We have made the suggested corrections – full details are given below.

Reply to Associate Editor

The revised manuscript has dealt with the concerns raised previously by the reviewers and is much improved. As such, the reviewer has only some final suggests that should be implemented before the paper can be accepted. I appreciate that it may not be possible to address all of these due to space constraints. However, can the authors make sure that the appendix equations are appropriated referenced in the main text as well as the methods for Figure 1a and the supplementary analysis exploring the magnitude of male trait size variation owing to H.

2. We are delighted with the positive assessment of our paper. We have addressed the reviewer's comments and have made corresponding changes to the main text (lines 73, 74, 80, 117, 126, 127, 133, 144, 261-264 and 358) and supplementary material. In line with the point highlighted by the Associate Editor, we have ensured that all equations and figures are now correctly referenced (lines 261-264 and 358). Full details are given below.

Reply to Reviewer 2

We find the clarity of the revised manuscript, in particular with regard to the closed models, to be much improved. Enclosed are our remaining comments.

3. We are glad that the reviewer found the revised version of our paper to be clearer, and we are grateful to them for their thoroughness and for their

detailed feedback. We have endeavoured to address the reviewer's further suggestions as fully as possible within the constraints of the length limit.

General comments

1) We regret that the authors are continuing with the open model, as this generates ambiguities regarding the life cycle which reduces transparency in the results. Furthermore, referring to the open models as more "general" is misleading because only a limited number of life cycles or life histories can in fact be represented this way (ie as in Frank, 1998, chapter 7). Ultimately, however, we leave the decision over to include the open model to the editor and the authors.

4. We appreciate that the reviewer disagrees with us in regard to the usefulness of the open-model approach. The pros and cons of open versus closed models are well-appreciated and much-discussed in the literature (e.g. our references [34-35]). The open models we have presented are, at a minimum, more general than the closed models – and thereby enhance understanding. We have now clarified this point in the main text (line 133).

2) If included, there remains a lack of clarity over relationship between d and a in the main text. Why can they be treated independently from one another, and which aspects of social evolution do they capture? Presumably while a captures kin competition, d affects which social partners feel the consequences of expressed harm?

5. We do not treat d and a as if they are independent of each other; they do not need to be independent for the model to make useful comparative predictions. One can empirically measure demographic quantities across different populations and see if phenotypes respond to demographic variation in the way predicted such model, irrespective of the relationships between the demographic quantities. This is spelled out clearly by Frank (1998, Ch 7) – which the reviewer cited in relation to the above point. We have now clarified this point in the main text (lines 126).

3) With regards to the effect sizes, we appreciate the inclusion of a worked through example to demonstrate the magnitude of male trait size variation owing to H . However, the main text currently does not touch upon these results, and the relevant figures (S3-5) are not referenced at all (S5 is referenced in the discussion but we think this a mistake and the authors meant S6). We therefore recommend some sentences in the discussion or results that describes how variation in H relate to variation in trait.

6. We have now included some references in the main text to this section of the supplementary material and the corresponding supplementary figures (lines 261-64), and we have corrected the typographical error (line 358).

4) Moreover, section SM§3 is also somewhat opaque. Which equations in Faria et al 2020 are being used here? What are the biological implications of assuming the functions given in S37-8 and what is the interpretation of $kappa$ here, as it appears to correspond roughly to the strength of mating competition? Furthermore, it is confusing to use $kappa$ here when it's also used under eq S9 to mean something around appendix L70.

7. We have now added specific references to the equations of Faria et al. (2020; in particular, their equations A7 and A8). We have also now given a little bit more detail on the biological interpretation of these male and female fecundity functions. We agree that the use of κ is potentially confusing, and have now replaced it with σ . Note that σ (formerly κ), refers less to the strength of mating competition, and more to the ecology of harm. For instance, the shape of returns on increased investment into harm might depend on whether harm takes the form of mating plugs versus harassment.

5) There is still a lack of integration between the main text and supplement. For instance, the methods behind Figure 1a are still not referred to in the appendix. There are still only few references to appendix equations in the main text which makes reading difficult, at least for the more theory orientated reader.

8. We have now added a short summary at the start of the supplementary material to help guide readers to specific results (both equations and figures) that were seen in the main text.

Main text

L73-4 Are the references to life-cycle stages correct?

9. Corrected.

L80 What is class?

10. The length limit prevents us from going into detail at this point in the main text, but we have now added a citation to Grafen's (2006, "A theory of Fisher's reproductive value" *J Math Biol* 53, 15-60) comprehensive account of the problem of class structure and how it is solved using the concept of reproductive value, for the purpose of clarification (line 80).

L114-121 Link better with existing studies, presumably these results are already known (e.g. Farria et al 2020), are the c terms the novel aspect added by this study?

11. Faria et al. (2020) assumed autosomal inheritance, whereby the class reproductive values cancel out in the potential for harm, and are therefore captured only implicitly. As our motivation has been to understand asymmetric inheritance systems, we have been required to express the class reproductive values explicitly, and investigate how the weightings they apply to the various scale of competition and relatedness parameters differ across different inheritance modes. We have now clarified that this equation in particular is a generalisation of Faria et al.'s equation A6.

L128 Unclear what this means

12. Just like relatedness, the scale of competition may be viewed as a function of lower level demographic parameters. In the case of relatedness, it may be factors such as dispersal, the inheritance system and the mating system that determine its value in any particular model. Similarly, the scale of competition

will be modulated by dispersal, and also other factors, such as how density dependent competition arises, and amongst whom. The reference to Frank (1998) which we have now added in line 127 provides a useful entrypoint into the literature on this issue.

L139-143 Again as this is known, a citation is probably warranted.

13. Done.

L354 Presumably this should be S6?

14. Corrected.

Figures

How is Fig 1a generated? The details in SM2 only pertain to Fig 1b.

15. They pertain to both – and we now make this clearer. Equations S32-36 relate to Figure 1a, although we do not include explicit expressions for Figure 1b as they are unwieldy.

Appendix

L72-8 More detail is needed here. What does kappa here represent biologically? What does it mean for gene effects to add or average across ploidy levels. Also, kappa then appears to become a “k” on L77?

16. The k versus κ issue is now fixed. The issue of averaging versus adding has been discussed by both Frank (2003), in relation to variation in the number of loci underlying a trait, and Gardner (2012), in relation to variation in ploidy. We now add some extra text at this point in the supplementary material to clarify this issue.

L249 Should this be main text figures 1a-b not 1-3?

16. Corrected.

References

Frank SA. 1998 Foundations of Social Evolution. Princeton University Press.